# FLIP: A Provable Defense Framework for Backdoor Mitigation in Federated Learning

**Kaiyuan Zhang, Guanhong Tao, Qiuling Xu, Siyuan Cheng, Shengwei An,**
**Yingqi Liu, Shiwei Feng, Guangyu Shen, Pin-Yu Chen[†], Shiqing Ma[‡], Xiangyu Zhang**
Purdue University, [†]IBM Research, [‡]Rutgers University
{zhan4057, taog, xu1230, cheng535, an93}@cs.purdue.edu,
{liu1751, feng292, shen447, xyzhang}@cs.purdue.edu,
[†]pin-yu.chen@ibm.com, [‡]sm2283@cs.rutgers.edu

## ABSTRACT

Federated Learning (FL) is a distributed learning paradigm that enables different parties to train a model together for high quality and strong privacy protection. In this scenario, individual participants may get compromised and perform backdoor attacks by poisoning the data (or gradients). Existing work on robust aggregation and certified FL robustness does not study how hardening benign clients can affect the global model (and the malicious clients). In this work, we theoretically analyze the connection among cross-entropy loss, attack success rate, and clean accuracy in this setting. Moreover, we propose a trigger reverse engineering based defense and show that our method can achieve robustness improvement with guarantee (i.e., reducing the attack success rate) without affecting benign accuracy. We conduct comprehensive experiments across different datasets and attack settings. Our results on nine competing SOTA defense methods show the empirical superiority of our method on both single-shot and continuous FL backdoor attacks. Code is available at https://github.com/KaiyuanZh/FLIP.

## 1 INTRODUCTION

Federated Learning (FL) is a distributed learning paradigm with many applications, such as next word prediction (McMahan et al., 2017), credit prediction (Cheng et al., 2021a), and IoT device aggregation (Samarakoon et al., 2018). FL promises scalability and privacy as its training is distributed to many clients. Due to the decentralized nature of FL, recent studies demonstrate that individual participants may be compromised and become susceptible to backdoor attacks (Bagdasaryan et al., 2020; Bhagoji et al., 2019; Xie et al., 2019; Wang et al., 2020a; Sun et al., 2019). Backdoor attacks aim to make any inputs stamped with a specific pattern misclassified to a target label. Backdoors are hence becoming a prominent security threat to the real-world deployment of federated learning.

**Deficiencies of Existing Defense.** Existing FL defense methods mainly fall into two categories, robust aggregation (Fung et al., 2020; Pillutla et al., 2022; Fung et al., 2020; Blanchard et al., 2017; El Mhamdi et al., 2018; Chen et al., 2017) which detects and rejects malicious weights, and certified defense (Cohen et al., 2019; Xiang et al., 2021; Levine & Feizi, 2020; Panda et al., 2022; Cao et al., 2021) which provides robustness certification in the presence of backdoors with limited magnitude. Some of them need a large number of clean samples in the global server (Lin et al., 2020b; Li et al., 2020a), which violates the essence of FL. Others require inspecting model weights (Aramoon et al., 2021), which may cause information leakage of local clients. Existing model inversion techniques (Fredrikson et al., 2015; Ganju et al., 2018; An et al., 2022) have shown the feasibility of exploiting model weights for privacy gains. Besides, existing defense methods based on weights clustering (Blanchard et al., 2017; Nguyen et al., 2021) either reject benign weights, causing degradation on model training performance, or accept malicious weights, leaving backdoor effective. According to our results in the experiment section, the majority of existing methods only work in the single-shot attack setting where only a small set of adversaries participate in a few rounds and fall short in the stronger and stealthier continuous attack setting where the attackers continuously participate in the entire FL training.

**FLIP.** In this paper, we propose a **F**ederated **L**earn**I**ng **P**rovable defense framework (**FLIP**) that provides theoretical guarantees. For each benign local client, FLIP adversarially trains the local

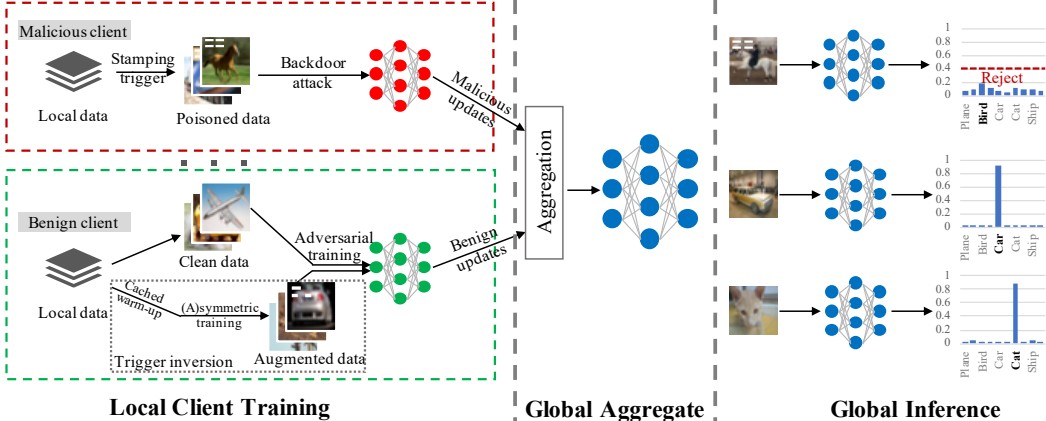

**Local Client Training** | **Global Aggregate** | **Global Inference**

Figure 1: Overview of FLIP. The left upper part (red box) performs the malicious client backdoor attack and the left lower part (green box) illustrates the main steps of benign client model training, they will submit local clients' updates to the global server. The middle part illustrates that the global server will aggregate all the received local clients' model weights and update the global server's model. The right part shows global server inference based on the updated global model. On benign clients, we do not assume any knowledge about the ground truth trigger.

model on generated backdoor triggers that can cause misclassification, which counters the data poisoning by malicious local clients. When all local weights are aggregated in the global server, the injected backdoor features in the aggregated global model are mitigated by the hardening performed on the benign clients. Therefore, FLIP can reduce the attack success rate of backdoor samples. The overview of FLIP is shown in Figure 1. As a part of the framework, we provide a theoretical analysis of how our training on a benign client can affect a malicious local client as well as the global model. To the best of our knowledge, this has not been studied in the literature. The theoretical analysis determines that our method ensures a deterministic loss elevation on backdoor samples with only slight loss variation on clean samples. It guarantees that the attack success rate will decrease, and the model can meanwhile maintain the main task accuracy on clean data without much degradation. Certified accuracy is commonly used in evasion attacks that do not involve training. As data poisoning happens during training, it is more reasonable to certify the behavior of models during training rather than inference.

**Our Contributions.** We make contributions on both the theoretical and the empirical fronts.

- We propose **FLIP**, a new provable defense framework that can provide a sufficient condition on the quality of trigger recovery such that the proposed defense is provably effective in mitigating backdoor attacks.

- We propose a new perspective of formally quantifying the loss changes, with and without defense, for both clean and backdoor data.

- We empirically evaluate the effectiveness of FLIP at scale across MNIST, Fashion-MNIST and CIFAR-10, using non-linear neural networks. The results show that FLIP significantly outperforms SOTAs on the continuous FL backdoor attack setting. The ASRs after applying SOTA defense techniques are still 100% in most cases, whereas FLIP can reduce ASRs to around 15%.

- We design an adaptive attack that is aware of the proposed defense and show that FLIP stays effective.

- We conduct ablation studies on individual components of FLIP and validate FLIP is generally effective with various downstream trigger inversion techniques.

**Threat Model.** We consider FL backdoor attacks performed by malicious local clients, which manipulate local models by training with poisoned samples. On benign clients, we do not assume any knowledge about the ground truth trigger. Backdoor triggers are inverted on benign clients based on received model weights (from the global server) and their local data (*non-i.i.d.*). Standard training on clean data and adversarial training on augmented data (clean samples stamped with inverted triggers) are then performed. The global server does not distinguish weights from trusted or untrusted clients. Nor does it assume any local data. Thus there is no information leakage or privacy violation.

The attack's goal is to inject a backdoor to the global model, achieving a high attack success rate without causing any noticeable model accuracy on clean samples. In our setting, a defender has no control over any malicious client, who may perform any kind of attack, e.g. model replacement or weight scaling. They can attack any round of FL. In an extreme case, they attack in every round after the global model converges (if an attack is persistent since the first round, the model may not converge (Xie et al., 2019)). In this paper, we consider static backdoors, i.e. patch backdoors (Gu et al., 2017). Dynamic backdoors such as reflection backdoors (Liu et al., 2020), composite backdoors (Lin et al., 2020a), and feature space backdoors (Cheng et al., 2021b) will be our future work.

## 2 RELATED WORK

**Backdoor Attack and Defense.** In general, the goal of backdoor attack is to inject a trigger pattern and associate it with a target label, e.g., by poisoning the training dataset. During testing, any inputs with such pattern will be classified as the target label. There are a number of existing backdoor attacks, like patch attacks (Gu et al., 2017; Liu et al., 2018), feature space attacks (Cheng et al., 2021b), etc. To identify whether a model is poisoned, existing works inverse triggers (Wang et al., 2019; Liu et al., 2019; Shen et al., 2021; Liu et al., 2022; Tao et al., 2022; Cheng et al., 2023), identify differences between clean models and backdoored models (Huang et al., 2020; Wang et al., 2020b). There are also methods that detect and reject inputs stamped with triggers (Ma & Liu, 2019; Li et al., 2020b).

**Federated Learning Backdoor Attack and Defense.** Federated learning distributes model training to multiple local clients and iteratively aggregates the local models to a shared global model. Since FL local model training is private, attackers could hijack some local clients and inject backdoor into the aggregated gobal model (Xie et al., 2019; Tolpegin et al., 2020; Bagdasaryan et al., 2020; Fang et al., 2020; Shejwalkar et al., 2022). To defend against FL backdoor attacks, a number of defense methods have been proposed (Blanchard et al., 2017; Pillutla et al., 2022; Sun et al., 2019; Nguyen et al., 2021; Ozdayi et al., 2020; Fung et al., 2020; Andreina et al., 2021; Cao et al., 2020), they focus more on detecting and rejecting malicious weights, which could fall short in the stronger and stealthier attack settings and may lead to data leakage. Certified and provable defense techniques (Panda et al., 2022; Cao et al., 2021; Xie et al., 2021) also have been proposed to analyze the robustness of FL, while they can provide robustness certification in the presence of backdoors with a (relatively) limited magnitude.

## 3 METHODOLOGY

In this section, we detail the design of FLIP, which consists of three main steps as illustrated in Figure 1. The procedure is summarized in Algorithm 1. (1) Trigger inversion. During local client training, benign local clients apply trigger inversion techniques to recover the triggers, stamp them on clean images (without changing their original labels) to constitute the augmented dataset. (2) Model hardening. Benign local clients combine the augmented data with the clean data to perform model hardening (adversarial training). The local clients submit updated local model weights to the global server, which aggregates all the received weights. (3) Low-confidence sample rejection. Our adversarial training can substantially reduce the prediction confidence of backdoor samples. During inference, we apply an additional sample filtering step, in which we use a threshold to preclude samples with low prediction confidence. Note that this filtering step is infeasible for most existing techniques as they focus on rejecting abnormal weights during training.

**Trigger Inversion.** Trigger inversion leverages optimization methods to invert the smallest input pattern that flips the classification results of a set of clean images to a target class. Neural Cleanse (Wang et al., 2019) uses optimization to derive a trigger for each class and eventually observes if there is any trigger that is exceptionally small and hence likely injected instead of naturally occurring feature. In our paper, we leverage *universal trigger inversion* that aims to generate a trigger that can flip samples of all the classes (other than the target class) to the target class.

**Class Distance.** Recent work quantifies model robustness (against backdoor) by *class distance* (Tao et al., 2022). Given images from a source class $s$, it generates a trigger, consisting of a mask $m$ and a pattern $\delta$, which can flip the labels of these images stamped with the trigger to a target class $t$. The

stamping function is illustrated in Equation 1 and the optimization goal in Equation 2, where $\mathcal{L}(\cdot)$ is the cross entropy loss, $M$ denotes the subject model, and $||\cdot||$ denotes $L^1$, i.e. the absolute value sum.

$$x'_{s \to t} = (1 - m) \cdot x_s + m \cdot \delta \qquad (1) \qquad Loss = \mathcal{L}(M(x'_{s \to t}), y_t) + \alpha \cdot ||m|| \qquad (2)$$

The class distance $d_{s \to t}$ is measured as $||m||$. The intuition here is that if it's easy to generate a small trigger from the source class to the target class, the distance between the two class is small. Otherwise, the class distance is large. Furthermore, the model is robust if all the class distances are large, or one can easily generate a small trigger between the two classes.

**Cached Warm-up.** Adversarial training on samples with inverted triggers is a widely used technique for model hardening. Observe that different label pairs have different distance capacities and enlarging label pair distances by model hardening can improve model robustness and help mitigate backdoors (Tao et al., 2022). Existing trigger inversion methods optimize all combinations of label pairs without selection and hence lead to quadratic computation time ($O(n^2)$). In order to reduce the trigger optimization cost, we first generate universal triggers, having each label being the target and prioritize promising pairs. It has a linear time complexity ($O(n)$). We consider pairs with larger *distance capacity* as having greater potential. Specifically, we need to know the difficulty of flipping a source class to the target class, we leverage loss changes during optimization to measure. In other words, classes far from the target class have larger loss variances as they are quite different from the target class; once the predicted label is flipped to the target class, the loss value would be quite small. During optimization, we compute the variance of loss between source classes and the target class. FLIP starts with a warm-up phase and repeats the above process for each class, and utilizes the loss changes of different source labels to approximate class distances. Such information is saved in a distance matrix or cache matrix (on each client). FLIP then prioritizes the promising pairs, namely, those with large distances. When the client is selected for hardening, FLIP generates the label-specific trigger for each source class and updates the distance matrix between the source class and target class.

---

**Algorithm 1** FLIP

---

1: **Globals input:** initial model parameters $w_0$, total training round $R_d$, randomly select a set of clients $K$
2: **Local client's input:** local dataset $D : \{x, y\}$ and learning rate $\eta$
3: **for** each training round $r$ in $[1, R_d]$ **do**
4:     **for** each client $k$ in $K$ **do**
5:         $w_{r+1}^k \leftarrow$ Local_Update $(w_r, D_k)$ ▷ The aggregator sends $w_r$ to client $k$ who inverts triggers based on $w_r$ and its data $D_k : D : \{x_k, y_k\}$ locally, and sends $w_{r+1}^k$ back to the aggregator
6:         $w_{r+1} \leftarrow \eta_r \sum_{k=1}^N w_{r+1}^k$ ▷ Global server aggregates all the received weights from the different clients
7: **Global output:** Global_Model_Inference $(w_{r+1}^k, \tau, x)$ ▷ Perform global model inference, $\tau$ is the confidence threshold, $x$ is the input
8: **function** LOCAL_UPDATE$(w_r, D_k)$
9:     **if** client $k$ was never selected **then**
10:         **for** each $s$ label existing in client $k$ **do** ▷ $s$ is source label, $t$ is target label
11:             distance_matrix$[s][t] \leftarrow L^1(s, t)$ ▷ Store all pair-wise class distances to a cache matrix
12:             promising_pairs $\leftarrow$ select(distance_matrix) ▷ Select top promising pairs from cache matrix
13:     **else if** client $k$ was selected before **then**
14:         promising_pairs $\leftarrow$ select(distance_matrix)
15:     **if** promising_pairs exist in dataset **then**
16:         $x_{adv} \leftarrow$ Symmetric_Inversion $(w_r, x_k, y_k)$
17:     **else**
18:         $x_{adv} \leftarrow$ Asymmetric_Inversion $(w_r, x_k, y_k)$
19:     $w_{r+1}^k \leftarrow$ Adversarial_Train $(\{x_k, x_{adv}\}, \{y_k, y_{adv}\})$ ▷ Adversarial training on clean and augmented data, $y_{adv}$ is the ground truth label of $x_{adv}$.
20:     return $w_{r+1}^k$

---

In Algorithm 1, each local client utilizes their local samples and computes their class distances (measured by $L^1$) to a target class and caches the results in the distance matrix (lines 11). Then based on the distance matrix, FLIP prioritizes the promising pairs with large distance (line 12). If the client has been selected before, we can directly get the promising pairs from the cached distance matrix (lines 13, 14). The distance matrix is stored and updated locally by each client. Caching allows more iterations allocated for model hardening.

**(A)symmetric Hardening.** Given a pair of *label 1* and *label 2*, there are two directions for trigger inversion, from *1* to *2* and from *2* to *1*. A straightforward idea (Tao et al., 2022) is to invert in both directions. However, it is impossible due to the *non-i.i.d* nature of client data in federated learning, namely, a local client's training data can be extremely unbalanced and there may be very few or even no samples for certain labels in a client. During model hardening (adversarial training), a promising class pair is selected for hardening in each iteration, according to the distance matrix mentioned above. We hence separate the model hardening into bidirectional or single directional based on data availability, accordingly symmetric or asymmetric model hardening. That is, if there exist sufficient data for the two labels of a class pair, symmetric hardening is carried out by generating triggers for the two directions and stamped on the corresponding source samples simultaneously (Alg 1 lines 15, 16). If there are only samples of one label of the pair, we then only harden the direction from the label to the target (Alg 1 lines 17, 18). Recovered trigger examples are shown in Appendix A.1 Figure 2 (c). After benign clients finish model hardening, they submit the updated model weights to the global server (Alg 1 lines 5, 19, 20), then the global server aggregates all the received client weights (Alg 1 line 6) and performs the global inference (Alg 1 line 7).

---

**Algorithm 2** Symmetric and Asymmetric Inversion

---

1: **Input:** local model parameters $W_r$, local client data $\{\mathbf{x}_k, y\}$
2: **Initialization:** $X_n \leftarrow$ a batch of $x \in \mathbf{x}_k$
3: **Initialization:** Initialize model $M$ from model weights $W_r$, $(m_{init}, \delta_{init})$, label $(a, b)$, indicator vector $\mathbf{p}$
4: **function** SYMMETRIC_INVERSION($W_r, \mathbf{x}_k, y$)
5:     $m, \delta \leftarrow$ Trigger_Initial $(m_{init}, \delta_{init})$
6:     **for** $step$ **in** $[0, max\_steps]$ **do**
7:         $X'_n = \mathbf{p} \cdot \big((1 - m[0]) \cdot X_n + m[0] \cdot \delta[0]\big) + (1 - \mathbf{p}) \cdot \big((1 - m[1]) \cdot X_n + m[1] \cdot \delta[1]\big)$ ▷ Indices of 0 and 1 denote optimization direction, $\mathbf{p}$ denotes the direction of symmetric hardening
8:         distance_matrix[a][b] $\leftarrow L^1$(a, b)
9:         distance_matrix[b][a] $\leftarrow L^1$(b, a)                     ▷ Update distance matrix in both directions
10:     return $X'_n$
11: **function** ASYMMETRIC_INVERSION($W_r, \mathbf{x}_k, y$)
12:     $m, \delta \leftarrow$ Trigger_Initial $(m_{init}, \delta_{init})$
13:     **for** $step$ **in** $[0, max\_steps]$ **do**
14:         $X'_n = (1 - m) \cdot X_n + m \cdot \delta$
15:     distance_matrix[a][b] $\leftarrow L^1$(a, b)                     ▷ Update distance matrix in a single direction
16:     return $X'_n$
17: **function** TRIGGER_INITIAL($m_{init}, \delta_{init}$)
18:     **if** $m_{init}$ is not None and $\delta_{init}$ is not None **then**
19:         $m, \delta \leftarrow m_{init}, \delta_{init}$                     ▷ Initialize mask and backdoor variables
20:     **else**
21:         $m, \delta \leftarrow$ random init with shape of $x \in \mathbf{x}_k$
22:     return $m, \delta$

---

In Algorithm 2, we present more details about the symmetric and asymmetric inversion. If the client has sufficient data (i.e. more than 5 images) for both labels of a pair $(a, b)$, we perform symmetric hardening by generating triggers from two directions ($a \rightarrow b$ and $b \rightarrow a$) simultaneously. We first initialize the backdoor mask $m$ and patter $\delta$ from two directions (line 5). The indicator vector $p$ denotes the direction of symmetric hardening, i.e. 1 denotes from label $a$ to $b$ and 0 denotes from $b$ to $a$. Then we stamp the triggers on the corresponding class samples (line 7), and update the distance matrix from two directions (line 8, 9). If the client only has sufficient data on the source label of a pair $(a, b)$, we perform asymmetric hardening from the source label to the target (i.e., $a \rightarrow b$). We initialize backdoor mask $m$ and patter $\delta$ in one direction (line 12), then stamp the triggers on the corresponding class samples (line 14), and update the distance matrix in one direction (line 15).

**Low-confidence Sample Rejection.** As the hardening on benign clients counters the data poisoning on malicious clients, the aggregated model on the global server tends to have low confidence in predicting backdoor samples (while the confidence on benign samples is largely intact). During the inference of global model, we apply a threshold $\tau$ to filter out samples with low prediction confidence after the softmax layer, which significantly improves the model's robustness against backdoor attacks in federated learning. In the next section, we prove that, as long as the inverted trigger satisfies our given bound, we can guarantee attack success rate must decrease and in the meantime the model can maintain similar accuracy on clean data.

# 4 THEORETICAL ANALYSIS

In this section, we develop a theoretical analysis to study the effectiveness of our proposed defense in a simple but representative FL setting. It consists of the following: (i) developing upper and lower bounds quantifying the cross-entropy loss changes on backdoored and clean data in the settings of with and without the defense (Theorem 1); (ii) showing a sufficient condition on the quality of trigger recovery such that the proposed defense is provably effective in mitigating backdoor attacks (Theorem 2); (iii) following (ii), we show that inference with confidence thresholding on models trained with our proposed defense can provably reduce the backdoor attack success rate while maintaining similar accuracy on clean data. During analysis, we leverage inequality (Mitrinovic & Vasić, 1970) and use variable substitution to get the upper-bound and lower-bound. To the best of our knowledge, our analysis is new to this field due to the novel modeling of trigger recovery and model hardening schemes in our proposed defense method. These findings are also consistent with our empirical results in more complex settings.

**Learning Objective and Setting.** Suppose the $k$-th device holds the $n_k$ training samples: $\{\mathbf{x}_{k,1}, \mathbf{x}_{k,2}, \cdots, \mathbf{x}_{k,n_k}\}$, where $\mathbf{x} \in \mathbb{R}^{1 \times d_x}$. The model has one layer of weights $W \in \mathbb{R}^{d_x \times I}$. Label $\mathbf{q} \in \mathbb{R}^{1 \times I}$ is a one-hot vector for $I$ classes. In this work, we consider the following distributed optimization problem: $\min_W \left\{ F(W) = \sum_{k=1}^N g_k F_k(W) \right\}$ where $N$ is the number of devices, $g_k$ is the weight of the $k$-th device such that $g_k \geq 0$ and $\sum_{k=1}^N g_k = 1$, and $F(\cdot)$ the objective function. The local objective $F_k(\cdot)$ is defined by $F_k(W) = \frac{1}{n_k} \sum_{j=1}^{n_k} \mathcal{L}(W; \mathbf{x}_{k,j})$, where $\mathcal{L}(\cdot; \cdot)$ is loss function. In the global server, we can write the global softmax cross-entropy loss function as $\mathcal{L}_{global} = -\sum_{i=1}^I q_i \cdot logsoftmax(\mathbf{x}W)_i = -\sum_{i=1}^I q_i \cdot log(\frac{e^{(\mathbf{x}W)_i}}{\sum_{t=1}^I e^{(\mathbf{x}W)_t}}) = -\sum_{i=1}^I q_i(\mathbf{x}W)_i + log(\sum_{t=1}^I e^{(\mathbf{x}W)_t})$, $i$ and $t$ as label index for the $I$ classes. Appendix A.16 describes all used symbols in our paper.

In the theoretical analysis, we assume that we are under the FedAvg protocol (McMahan et al., 2017). In order to simplify analysis without losing generality, we assume there is one global server and two clients: one benign and the other malicious. We conduct the analysis on multi-class classification using logistic regression. The model is composed of one linear layer with the softmax function and the cross-entropy loss.

Theorem 1 develops the upper and lower bounds quantifying the loss changes on backdoored and clean data in the settings with and without the defense.

**Theorem 1** (Bounds on Loss Changes). *Let $\mathcal{L}'_g$ denote the global model loss with defense, $\mathcal{L}_g$ without defense, let $\Delta W = W' - W$ denote the weight differences with and without defense. The loss difference with and without defense can be upper and lower bounded by*

$$\min_t(\mathbf{x}\Delta W)_t - \sum_{i=1}^I q_i(\mathbf{x}\Delta W)_i \leq \mathcal{L}'_g - \mathcal{L}_g \leq \max_t(\mathbf{x}\Delta W)_t - \sum_{i=1}^I q_i(\mathbf{x}\Delta W)_i \qquad (3)$$

The detailed proof is provided in Appendix A.14. The above theorem bounds the loss changes with and without the defense. From Cauchy–Schwarz inequality (Mitrinovic & Vasić, 1970), we derive the lower bound and upper bounds of loss changes. Intuitively, given the parameter $W$ of the linear layer, the $k$-th device holds $n_k$ training samples $\mathbf{x}_{k,n_k}$, the loss differences of with and without defense must be larger than the lower bound $\Delta \min\_loss$ as $\min_t(\mathbf{x}_{k,n_k}\Delta W)_t - \sum_{i=1}^I q_i(\mathbf{x}_{k,n_k}\Delta W)_i$, and smaller than the upper bound $\Delta \max\_loss$ as $\max_t(\mathbf{x}_{k,n_k}\Delta W)_t - \sum_{i=1}^I q_i(\mathbf{x}_{k,n_k}\Delta W)_i$.

To facilitate the analysis of attack success rate (ASR) and clean accuracy (ACC) changes, intuitively, we aim to analyze how much the ASR at least will be reduced and how much the ACC will at most be maintained. Thus, we studied this lower bound ($\Delta \min\_loss$) on backdoor data, which indicates the minimal improvements on the backdoor defense that reduce the ASR. Similarly we studied the upper bound ($\Delta \max\_loss$) for clean data, as they indicates the worst-case accuracy degradation. Denote backdoor samples as $n_b$ and clean samples as $n_c$. Note that backdoor samples can be written as $\mathbf{x}_s + \delta$. By using Theorem 1, we have $\Delta \min\_loss = \sum_{s=1}^{n_b} \min_t[(\mathbf{x}_s + \delta)\Delta W]_t - \sum_{s=1}^{n_b} \sum_{i=1}^I q_{s,i}[(\mathbf{x}_s + \delta)\Delta W]_i$. And similarly on benign data, we have $\Delta \max\_loss$ $\mathbf{x}_s$, $\Delta \max\_loss = \sum_{s=1}^{n_c} \max_t(\mathbf{x}_s\Delta W)_t - \sum_{s=1}^{n_c} \sum_{i=1}^I q_{s,i}(\mathbf{x}_s\Delta W)_i$.

Next, we aim to develop a sufficient condition on the quality of trigger recovery such that the proposed defense is provably effective in mitigating backdoor attack and in the meantime maintaining similar accuracy on clean data, based on Theorem 1.

**Theorem 2** (General Robustness Condition). *Let $\alpha =$*

$$\frac{\eta_r \sum_{s=1}^{n_b} \sum_{i=1}^{I} (q_{s,i}^* - q_{s,i}) \{\mathbf{z_s} \sum_{j=1}^{n_1} [\mathbf{z_j}^T (\mathbf{q_j} - p(\mathbf{z_j}))]\}_i}{\mathbf{b} \left\{ \eta_r \sum_{s=1}^{n_b} \sum_{i=1}^{I} (q_{s,i}^* - q_{s,i}) \{\sum_{j=1}^{n_1} [\mathbf{z_j}^T (\mathbf{q_j} - p(\mathbf{z_j}))]\}_i \right\}}$$

*where $\mathbf{b} = [b_1, ..., b_d]$, $d$ is the sample dimension,*

*let $b_v = \text{sign} \left\{ \eta_r \sum_{s=1}^{n_b} \sum_{i=1}^{I} (q_{s,i}^* - q_{s,i}) \sum_{j=1}^{n_1} [\mathbf{z_j}^T \mathbf{q_j} - p(\mathbf{z_j})]]_{i,v} \right\}$, on all dimensions $v$ of the vector. For all $||\epsilon||_\infty \leq \alpha$, we have $\Delta \min\_loss \geq 0$.*

*And we have $\Delta \max\_loss \leq \eta_r \sum_{s=1}^{n_c} \sum_{i=1}^{I} (q_{s,i}^* - q_{s,i}) \mathbf{x_s} \sum_{j=1}^{n_1} [\mathbf{z_j}^T (\mathbf{q_j} - p(\mathbf{z_j}))]_i$.*

The detailed proof is provided in Appendix A.15. We denote $q_s^*$ as a one-hot vector for sample $s$ with $I$ dimensions. Its $i$-th dimension is defined as $q_{s,i}^*$, and $q_{s,i}^* = 1$ if $i = \arg\min_t [(\mathbf{x}_s + \delta)(W' - W)]_t$. Denote $\delta$ as ground truth trigger, $\epsilon$ as difference of reversed trigger and ground truth trigger, $\eta_r$ as the learning rate (a.k.a. step size) in round $r$. Denote $\mathbf{z}$ as $\mathbf{x} + \delta + \epsilon$ for simplicity, which is the benign sample stamped with the recovered trigger. Note that $\Delta \min\_loss \geq 0$ indicates that the defense is provably effective than without defense. Since benign local clients' training can increase global model backdoor loss, and they have positive effects on mitigating malicious poisoning effect, the second condition $\Delta \max\_loss \leq \eta_r \sum_{s=1}^{n_c} \sum_{i=1}^{I} (q_{s,i}^* - q_{s,i}) \mathbf{x_s} \sum_{j=1}^{n_1} [\mathbf{z_j}^T (\mathbf{q_j} - p(\mathbf{z_j}))]_i$ indicates that the defense is provably guarantee maintaining similar accuracy on clean data, similarly, here $i = argmax_t [\mathbf{x}_s (W' - W)]_t$ (with a slight abuse of the notation $q_s^*$).

**Corollary 1.** *Assume $\epsilon$ satisfies Theorem 2, let $n_b$ as backdoored samples, $n_c$ as benign samples, $\tau$ as confidence threshold. Then the number of backdoored samples that are rejected is $R_{bd} = R_b' - R_b$, the number of benign samples that are rejected is $R_{bn} = R_c' - R_c$*

$R_b'$ and $R_b$ denote the rejected backdoor samples *with* and *without* defense. With defense, $R_b'$ is $\sum_{j=1}^{n_b} \mathbf{1}(\mathcal{L}_g + \Delta \min\_loss > \mathcal{L}_\tau)$; and without defense, $R_b$ is $\sum_{j=1}^{n_b} \mathbf{1}(\mathcal{L}_g > \mathcal{L}_\tau)$. Thus, the exact value of rejected backdoored samples can be calculated through $R_{bd} = R_b' - R_b$.

Similarly, $R_c'$ and $R_c$ denote the rejected clean(benign) samples *with* and *without* defense. With defense, $R_c'$ is $\sum_{j=1}^{n_c} \mathbf{1}(\mathcal{L}_g + \Delta \max\_loss > \mathcal{L}_\tau)$; and without defense, $R_c$ is $\sum_{j=1}^{n_c} \mathbf{1}(\mathcal{L}_g > \mathcal{L}_\tau)$. Thus, the exact value of rejected benign samples can be calculated through $R_{bn} = R_c' - R_c$.

## 5 EXPERIMENT

In this section, we empirically evaluate FLIP under two existing attack settings, i.e. single-shot attack (Bagdasaryan et al., 2020) and continuous attack (Xie et al., 2019). We compare the performance of FLIP with 9 state-of-the-art defenses, i.e. Krum (Blanchard et al., 2017), Bulyan Krum (El Mhamdi et al., 2018), RFA (Pillutla et al., 2022), FoolsGold (Fung et al., 2020), Median (Yin et al., 2018), Trimmed Mean (Yin et al., 2018), Bulyan Trimmed Mean (Buly-Trim-M) (El Mhamdi et al., 2018), and FLTrust (Cao et al., 2020), DnC (Shejwalkar & Houmansadr, 2021). Besides using the experiment settings in (Bagdasaryan et al., 2020) and (Xie et al., 2019), we conduct experiments using the setting in our theoretical analysis to validate our analysis. Moreover, we evaluate FLIP on adaptive attacks and conduct several ablation studies.

### 5.1 EXPERIMENT SETUP

We conduct the experiments under the PyTorch framework (Paszke et al., 2019) and reimplement existing attacks and defenses following their original designs. Regarding the attack setting, there are 100 clients in total by default. In each round we randomly select 10 clients, including 4 adversaries and 6 benign clients. Following the existing works setup, we use SGD and trains for $E$ local epochs ($E$ is 5 in continuous attacks and 10 in single-shot attacks) with local learning rate $lr$ ($lr$ is 0.1 in benign training and 0.05 in poison training) and batch size 64. More setup details in Appendix A.1.

**Dataset.** Our experiments are conducted on 3 well-known image classification datasets, i.e. MNIST (LeCun et al., 1998), Fashion-MNIST (F-MNIST) (Xiao et al., 2017) and CIFAR-10 (Krizhevsky et al., 2009). Note that our data are not independent and identically distributed

(*non-i.i.d.*) which is more practical in real-world applications. We leverage the same setting as (Bagdasaryan et al., 2020) which applies a Dirichlet distribution (Minka, 2000) with a parameter $\alpha$ as 0.5 to model *non-i.i.d.* distribution. The poison ratio denotes the fraction of backdoored samples added in each training batch, MNIST and Fashion-MNIST poison ratio is 20/64, CIFAR-10 is 5/64.

**Evaluation Metrics.** We consider attack success rate (ASR) and main task accuracy (ACC) as evaluation metrics to measure defense effectiveness. ASR indicates the ratio of backdoored samples that are misclassified as the attack target label, while ACC indicates the ratio of correct classification on benign samples. While certified accuracy is commonly used in evasion attacks that do not involve training. As data poisoning happens during training, it is more reasonable to certify the behavior of models during training rather than inference.

## 5.2 Evaluation on Backdoor Mitigation

We consider backdoor attacks via model replacement approach where attackers train their local models with backdoored samples. We follow single-shot setting in (Bagdasaryan et al., 2020) and continuous setting in (Xie et al., 2019) to perform the attacks. Single-shot backdoor attack means every adversary only participates in one single round, while there can be multiple attackers. Continuous backdoor attack is more aggressive where the attackers are selected in every round and continuously participate in the FL training from the beginning to the end. Both settings of attacks happen after the global model converges, since if attackers poison from the first round, even after training enough rounds, (Xie et al., 2019) found that the main accuracy was still low and models hard to converge.

For fair comparison, we report ASR and ACC of FLIP and all the baselines in the same round. Note that the confidence threshold $\tau$ of FLIP introduced in Section 3 is only used in the continuous attacks to filter out low-confidence predictions, since single-shot attack is easy to defend. Based on our empirical study, we typically set $\tau = 0.3$ for MNIST and Fashion-MNIST, and $\tau = 0.4$ for CIFAR-10, and we evaluate thresholds with the Area Under the Curve metric (Appendix A.4). Single-shot attack results shown in Table 1. Line 2 illustrates the attack performance with *No Defense*. Observe that single-shot attack can achieve more than 80% ASR throughout all the datasets while preserving a high main task accuracy over 77%. The follow-

Table 1: Single-shot attack evaluation

| Baselines | MNIST | | F-MNIST | | CIFAR-10 | |
|---|---|---|---|---|---|---|
| | ACC | ASR | ACC | ASR | ACC | ASR |
| No Defense | 97.55 | 80.12 | 81.01 | 96.72 | 77.52 | 80.46 |
| Krum | 97.50 | 0.35 | 79.49 | 10.79 | 77.00 | 9.51 |
| Bulyan Krum | 97.76 | 0.39 | 81.45 | 6.42 | 79.65 | 5.77 |
| RFA | 97.93 | 0.39 | 81.82 | 4.39 | 79.54 | 6.13 |
| Trimmed Mean | 97.81 | 0.38 | 81.81 | 5.40 | 79.95 | 5.81 |
| Buly-Trim-M | 97.02 | 90.75 | 79.84 | 99.38 | 66.69 | 84.05 |
| FoolsGold | 97.51 | 0.39 | 80.59 | 5.64 | 78.67 | 3.70 |
| Median | 97.76 | 0.37 | 81.76 | 5.97 | 64.31 | 2.39 |
| FLTrust | 97.26 | 0.48 | 79.92 | 7.69 | **72.44** | **2.18** |
| DnC | 97.61 | 0.46 | 80.33 | 5.35 | 76.66 | 5.36 |
| FLIP | **96.05** | **0.13** | **78.20** | **3.16** | 73.41 | 7.83 |

ing rows show the defense performance of existing SOTA defenses and the last row denotes FLIP results. We can find that FLIP can reduce the ASR to below 8% on all the 3 datasets and keep the benign accuracy degradation within 5%. FLIP outperforms all the baselines on both MNIST and Fashion-MNIST while is slightly worse on CIFAR-10.

We show the results of continuous attack in Table 2. Continuous attack is more aggressive than the single-shot one. The former's ASR is 3%-20% higher than the latter when there is no defense. Note that all the existing defense techniques fail in the continuous attack setting. The ASR remains nearly 100% in most cases on MNIST and Fashion-MNIST and is higher than 63% on CIFAR-10. However, FLIP reduces the ASR to a low level and the accuracy degradation is within an acceptable range. Specifically, FLIP reduces the ASR on MNIST to 2% while the accuracy degradation is within 2%. For Fashion-MNIST and CIFAR-10, the ASR is reduced to below 18% and 23%, respectively, while the accuracy decreases a bit more

Table 2: Continuous attack evaluation

| Baselines | MNIST | | F-MNIST | | CIFAR-10 | |
|---|---|---|---|---|---|---|
| | ACC | ASR | ACC | ASR | ACC | ASR |
| No Defense | 98.71 | 100.00 | 80.35 | 99.99 | 77.83 | 84.73 |
| Krum | **97.59** | **0.14** | 73.18 | 20.03 | 40.29 | 18.79 |
| Bulyan Krum | 98.15 | 94.01 | 82.17 | 99.46 | 68.61 | 97.31 |
| RFA | 98.54 | 100.00 | 85.69 | 100.00 | 79.39 | 63.10 |
| Trimmed Mean | 98.52 | 100.00 | 84.59 | 99.99 | 75.18 | 91.84 |
| Buly-Trim-M | 98.80 | 100.00 | 76.18 | 99.93 | 71.91 | 68.83 |
| FoolsGold | 97.91 | 99.99 | 80.58 | 99.98 | 74.57 | 78.30 |
| Median | 98.14 | 66.01 | 84.07 | 99.34 | 57.01 | 69.99 |
| FLTrust | 91.96 | 20.60 | 74.63 | 35.36 | 74.85 | 68.70 |
| DnC | 89.81 | 99.78 | 65.90 | 97.73 | 53.16 | 82.37 |
| FLIP | 96.62 | 1.93 | **72.99** | **17.65** | **71.28** | **22.90** |

compared to the results of MNIST and to single-shot attack. This is reasonable due to the following: First, the complexity of the dataset and continuous backdoor attacks may add to the difficulty of recovering good quality triggers. In addition, there is trade-off between adversarial training accuracy and standard accuracy of a model as discussed in (Tsipras et al., 2019). Adversarial training on benign clients can induce negative effects on the accuracy. However, we argue that FLIP still outperforms existing defenses as the ASR is reduced to a low level.

### 5.3 EVALUATION ON THE SAME SETTING AS THEORETICAL ANALYSIS

In this section, we conduct an experiment that follows the same setting as our assumptions in the theoretical analysis to validate its correctness. We conduct experiments on the multi-class logistic regression (i.e., one linear layer, softmax function, and cross-entropy loss) as the setting in theoretical analysis Section 4. We take MNIST as the example for analysis and it can be easily extended to other datasets. Regarding the FL system setting, there are one global server, one benign client and one malicious client. We train the FL global model until convergence and then apply the attack. The attackers inject the pixel-pattern backdoor in images and swap the label of the image source label to the target label. We also do not have any restrictions on the attackers, as long as they follow the federated learning protocol.

Table 3 shows the result of single-shot and continuous attack ACC and ASR on the logistic regression. We can see both single-shot and continuous attacks' ASRs are reduced to around 5% and the accuracy degradations are within an acceptable range. This result is consistent with our observations on more complex settings above. Besides, the exact value of rejected clean samples and backdoored samples under different settings can also be calculated, which correspond to Corollary 1, details can be found in Appendix A.2.

Table 3: Logistic regression evaluation

| Attack Type | Metric | No Defense | FLIP |
|---|---|---|---|
| Single-Shot | ACC | 88.43 | 84.58 |
| | ASR | 64.48 | 5.28 |
| Continuous | ACC | 83.03 | 80.76 |
| | ASR | 63.78 | 4.90 |

### 5.4 ADAPTIVE ATTACKS

We study an attack scenario where the adversary has the knowledge of FLIP, our results show that FLIP still mitigate the backdoor attacks in most cases. For those that ACC does degrade, the adaptive attack is not effective. Details can be found in Appendix A.3.

### 5.5 ABLATION STUDY

This section we conduct several ablation studies. We study both adversarial training (Appendix A.5) and thresholding (Appendix A.6) is critical in FLIP. We study another different trigger inversion technique in FLIP, which can mitigate backdoors as well, indicating that FLIP is compatible with different trigger inversion techniques (Appendix A.7). We study the effects of different sizes of triggers and show that our defense can cause a significant ASR reduction while maintaining comparable benign classification performance (Appendix A.8). We also study different threshold influences on ACC and ASR and show the trade-off between attack success rate and accuracy (Appendix A.9).

## 6 CONCLUSION

We propose a new provable defense framework FLIP for backdoor mitigation in Federated Learning. The key insight is to combine trigger inversion techniques with FL training. As long as the inverted trigger satisfies our given bound, we can guarantee attack success rate will decrease and in the meantime the model can maintain similar accuracy on clean data. Our technique significantly outperforms prior work on the SOTA continuous FL backdoor attack. Our framework is general and can be instantiated with different trigger inversion techniques. While applying various trigger inversion techniques, FLIP may have slight accuracy degradation, but it can significantly boost the robustness against backdoor attacks.

ACKNOWLEDGEMENTS

We thank the anonymous reviewers for their constructive comments. This research was supported, in part by IARPA TrojAI W911NF-19-S-0012, NSF 1901242 and 1910300, ONR N000141712045, N000141410468 and N000141712947. Any opinions, findings, and conclusions in this paper are those of the authors only and do not necessarily reflect the views of our sponsors.

ETHICS STATEMENT

In this paper, our studies are not related to human subjects, practices to data set releases, discrimination/bias/fairness concerns, and also do not have legal compliance or research integrity issues. Backdoor attacks aim to make any inputs stamped with a specific pattern misclassified to a target label. Backdoors are hence becoming a prominent security threat to the real-world deployment of federated learning. FLIP is a provable defense framework that can provide a sufficient condition on the quality of trigger recovery such that the proposed defense is provably effective in mitigating backdoor attacks.

REPRODUCIBILITY STATEMENT

The implementation code is available at `https://github.com/KaiyuanZh/FLIP`. All datasets and code platform (PyTorch) we use are public. In addition, we also provide detailed experiment parameters in the Appendix.

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

## A   APPENDIX

We provide a simple table of contents below for easier navigation of the appendix.
**CONTENTS**

**Appendix A.1** Provides more details on experimental setups, neural network structures, parameters, etc.
**Appendix A.2** Provides extended evaluation on the same setting as theoretical analysis.
**Appendix A.3** Provides evaluation on adaptive attacks.
**Appendix A.4** Shows AUC-ROC curve of our confidence-based sample rejection on MNIST.
**Appendix A.5** Studies adversarial training can significantly reduce attackers' poisoning confidence.
**Appendix A.6** Validates thresholding is critical during evaluation.
**Appendix A.7** Provides evaluation on other trigger inversion technique.
**Appendix A.8** Analyzes the impact of different trigger sizes.
**Appendix A.9** Studies different threshold influences on ACC and ASR.
**Appendix A.10** Provides additional evaluation results on the existing defenses.
**Appendix A.11** Provides justifications for SOTA defenses not working.
**Appendix A.12** Analyzes the impact of trigger quality.
**Appendix A.13** Data Distribution Effects.
**Appendix A.14** Provides proofs for our Theorem 1.
**Appendix A.15** Provides proofs for our Theorem 2.
**Appendix A.16** Glossary of notations.

### A.1   EXPERIMENT SETUP

In this section, we illustrate more details about the experimental setups, neural network structures, parameters setups, etc. For more detailed hyperparameter settings and evaluations, please refer to our code repository, we will release our code upon the paper acceptance.

We train the FL system following our FLIP framework on three datasets: MNIST (LeCun et al., 1998), Fashion-MNIST (Xiao et al., 2017) and CIFAR-10 (Krizhevsky et al., 2009). MNIST has a training set of 60,000 examples, and a test set of 10,000 examples and 10 classes. Fashion-MNIST consists of a training set of 60,000 examples and a test set of 10,000 examples. Each example is a 28x28 grayscale image, associated with a label from 10 classes. CIFAR-10 is an object recognition dataset with 32x32 colour images in 10 classes. It consists of 60,000 images and is divided into a training set (50000 images) and a test set (10000 images). We split the training data for FL clients in a *non-i.i.d.* manner, by a Dirichlet distribution (Minka, 2000) with hyperparameter $\alpha$ 0.5, following the same setting as (Bagdasaryan et al., 2020; Xie et al., 2019). We train the FL global model until convergence and then apply various trigger inversion defense techniques, otherwise the main task accuracy is low and the backdoored model is hard to converge (Xie et al., 2019). In the augment dataset, we use 64 augmented samples in each batch of each local client training, following the existing works of backdoor removal in (Tao et al., 2022). Note that the confidence threshold $\tau$ of FLIP discussed in Methodology Section 3 is only used in continuous backdoor attack setting to filter out low-confidence predictions. Based on our empirical study, we typically set $\tau = 0.3$ for simpler datasets, e.g. MNIST and Fashion-MNIST, while $\tau = 0.4$ for more complex datasets, e.g. CIFAR-10. We apply two convolutional layers and two fully connected layers in MNIST and Fashion-MNIST, and Resnet-18 in CIFAR-10 to train our model.

**FLTrust Setting** In FLTrust (Cao et al., 2020), we follow the original settings and collect the root dataset for the learning task with 100 training examples, the root dataset has the same distribution as the overall training data distribution of the learning task. We exclude the sampled root dataset from the clients' local training data, indicating that the root dataset is collected independently by the global server.

**Class distance** The class distance is defined by the trigger size, we use $L^1$ norm to measure. The intuition here is that if it's easy to generate a small trigger from the source class to the target class, the distance between the two classes is small. Otherwise, the class distance is large. Furthermore, the model is robust when all the class distances are large, otherwise one can easily generate a small trigger between the two classes. The class distances of models in the wild are very small and do not align well with humans' intuition, for example, classes turtle and bird have smaller distances than

classes cat and dog, which makes the models vulnerable to backdoor attacks (Tao et al., 2022). The class distance is independent of the number of local client samples, in other words, the class distance is related to the model's robustness itself instead of the number of samples.

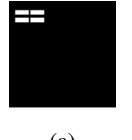 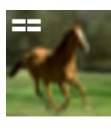 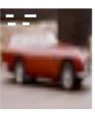

(a)                    (b)                    (c)

Figure 2: Trigger Examples, (a) is the ground truth trigger, (b) is malicious client poisoned data, (c) is after benign client trigger inversion, augmented data

Regarding the attack setting, there are 100 clients in total by default. In each round we randomly select 10 clients, including 4 adversaries and 6 benign clients. We do not have any restrictions on attackers as long as they follow the federated learning communication protocol. The attackers inject the pixel-pattern backdoor in images and swap the label of image source label to target label (by default, label "2"). Figure 2 (b) shows a backdoored example. During testing phase, any inputs with such pattern will be classified as the target label. In single-shot attack, attackers can choose any round to participate. In continuous attack, attackers participate in every round after model convergence. Benign clients perform adversarial training continuously in both settings. We report the ACC and ASR after the attack happens at least 60 rounds, that is, attackers already achieve a high and stable attack success rate.

## A.2 EVALUATION ON THE SAME SETTING AS THEORETICAL ANALYSIS

In this section, we conduct an extend experiment that follows the same setting as our assumptions to validate our theoretical analysis.

Table 4 shows the sample counts of clean samples and backdoored samples under different settings. As illustrated in Corollary 1, the exact value of rejected backdoored samples ($R_b/R_b'$) and rejected clean samples ($R_c/R_c'$) can be calculated. The *No defense* column represents the counts of $R_b$ and $R_c$ *without* defense; the last column *FLIP* represents the counts of $R_b'$ and $R_c'$ *with* defense, which correspond to the numbers defined in theoretical analysis.

Table 4: Logistic regression sample counts

| Attack Type | Samples count | Total samples | No defense | FLIP |
|---|---|---|---|---|
| Single-Shot | Clean | 10000 | 8843 | 8458 |
|  | Poisoned | 9020 | 5816 | 476 |
| Continuous | Clean | 10000 | 8303 | 8076 |
|  | Poisoned | 9020 | 5753 | 442 |

## A.3 RESILIENCE TO ADAPTIVE ATTACKS

As attackers may work out adaptive attacks to get over FLIP, in this section, we design a counter-measure for attackers and evaluate FLIP under the adaptive attack scenario. Detailed adaptive attack consists of the following steps: (1) attackers apply the same trigger inversion technique as benign clients to obtain the inverted triggers; (2) attackers stamp the inverted triggers to their local images and add them to the training phase for backdoor attacks; (3) attackers submit the updated model weights to global server. We conduct experiments on three datasets under continuous attack setting. Table 5 shows the result, observe that even under an adaptive attack setting, FLIP can still mitigate the backdoor attacks in both MNIST and Fashion-MNIST. In CIFAR-10, the accuracy drops and the adaptive attack is not effective. This indicates that even though the attackers are aware of our technique during poison training, under the FLIP framework, benign clients can still effectively reduce the attacker's poisoning confidence and keep the attack success rate in a low range.

Table 5: Adaptive attacks evaluation

| Continuous attack | ACC | ASR |
|---|---|---|
| MNIST | 96.82 | 0.61 |
| Fashion-MNIST | 73.42 | 18.94 |
| CIFAR-10 | 60.08 | 14.02 |

## A.4 AREA UNDER THE CURVE (AUC)

In this section, we take MNIST as an example to show the AUC-ROC curves (Area Under the Curve Receiver Operating Characteristics), other datasets can be analyzed similarly. Figure 3 shows the AUC-ROC curve of our confidence-based sample rejection on MNIST. The curve is plotted with TPR (True Positive Rate) against the FPR (False Positive Rate ) where TPR is on the $y$-axis and FPR is on the $x$-axis. In our evaluation, the AUC is 0.97. As stated by many existing works (Mandrekar, 2010), an AUC of 0.5 (indicated by the orange dashed line) means the model is unable to discriminate positive and negative samples while an AUC higher than 0.9 is considered outstanding. Therefore our confidence-based rejection strategy is effective in distinguishing backdoored samples and benign samples.

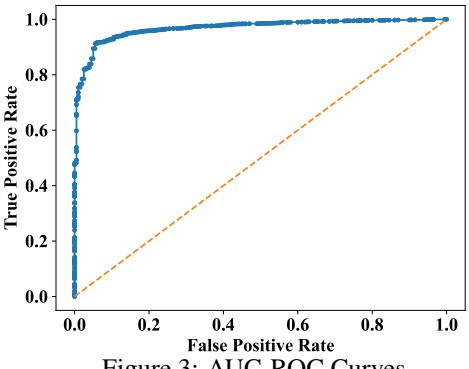

Figure 3: AUC-ROC Curves

## A.5 EFFECT OF ADVERSARIAL TRAINING

In this section, we aim to validate that adversarial training in benign local clients indeed can bring positive effects in reducing attackers' poisoning confidence. We conduct the experiments under continuous attacks. We use the same threshold $\tau$ and only remove adversarial training at benign local clients and keep all the other settings the same, e.g. confidence threshold.

Table 6: Effect of adversarial training

| Continuous attack | ACC | ASR |
|---|---|---|
| MNIST | 96.88 | 51.75 |
| Fashion-MNIST | 79.68 | 98.53 |
| CIFAR-10 | 72.90 | 83.13 |

In Table 6, we observe that without adversarial training, malicious can successfully inject backdoor patterns even with a high confidence threshold above $\tau$. The underlying reason is that adversarial training in benign clients hardens the model against malicious samples and reduces the confidence of malicious samples. Interestingly, we notice MNIST ASR drops compare with no defenses, the reason could be MNIST dataset feature is simpler, thus with a lot of benign clients continuous training parallelly, it is easy to forget injected backdoor patterns quickly (Wang et al., 2020a), so that attacker's poisoning confidence is reduced and parts samples are rejected. Hence, the results show that adversarial training is significantly effective in reducing the attacker's confidence in backdoor samples during backdoor training, which is consistent with our theoretical analysis.

## A.6 Effect of Confidence Threshold

In this section, we demonstrate that threshold is a critical component in FLIP and we evaluate our defense with and without thresholding, results can be found in Table 7. We conduct thresholding experiments under continuous backdoor attacks with three datasets. Each benign client performs trigger inversion and adversarial training as before, while in global inference-time, we set the confidence threshold $\tau$ to 0, which is no threshold, and keep all the other settings unchanged. We observe that without thresholding applied, though the ASRs are reduced to some extent, they are still much higher, compared with FLIP results in Table 2. The underlying reason is adversarial training does help in reducing the confidence of backdoored samples, however, without applying confidence threshold to reject the backdoored samples, ASR keeps high. We validate that the threshold is critical in FLIP and the observation is consistent with our results in Corollary 1.

Table 7: Effect of confidence threshold

| Continuous attack | ACC | ASR |
|---|---|---|
| MNIST | 97.20 | 22.35 |
| Fashion-MNIST | 78.76 | 30.67 |
| CIFAR-10 | 75.31 | 52.47 |

## A.7 Other Trigger Inversion Techniques Evaluation

In general, FLIP is compatible with any trigger inversion technique. In this section, we use another widely-used technique ABS (Liu et al., 2019) as the trigger inversion component of our framework. Specifically, we replace the "Trigger inversion" part in Figure 1 with ABS, while keeping all other settings the same. We conduct experiments on both single-shot and continuous attack settings. Note that we only evaluate CIFAR-10, since the released version of ABS focuses on the complex dataset with three color channels instead of greyscale images. In each training round of local clients, we use ABS to invert 10 most likely triggers and perform the adversarial training.

Table 8: Other Trigger Inversion Techniques Evaluation

| ABS | Single-shot | | Continuous | |
|---|---|---|---|---|
| | ACC | ASR | ACC | ASR |
| CIFAR-10 | 74.14 | 8.00 | 74.90 | 22.38 |

Table 8 shows the defense technique evaluation result, which is consistent with results shown previously in Table 1 and 2. We observe that in continuous attack, FLIP equipped with ABS keeps higher clean accuracy 74% compared to 71% in Table 2 and they both reduce ASR to a low level, near 22%. However, in the single-shot attack, FLIP with ABS only reduces ASR to 8%. The underlying reason is that ABS inverts effective triggers within a small size range, while the method in our main text is more aggressive in hardening the model. The result demonstrates that FLIP is generally effective with various downstream trigger inversion techniques against backdoor attacks.

## A.8 Impact of Trigger Size

In this section, we study the different sizes of triggers effect and the evaluation results in Table 9. We define the initial trigger size as X, that is, 2*X denotes the trigger size is scaled up two times compared with the initial trigger. Take MNIST as an example, we observe that the single-shot ASR is low when trigger size (TS) is 1*X, the reason is each local trigger is too small to be recognized during the global model testing phase. We conduct an experiment consisting of different trigger sizes from 1*X, 2*X, 4*X, 6*X, to 8*X. The evaluation shows that our defense can significantly degrade ASR while maintaining comparable benign classification performance, no matter how triggers' sizes change.

Table 9: Trigger Size

| TS | No Defense | | FLIP | |
|---|---|---|---|---|
| | ACC | ASR | ACC | ASR |
| 1* X | 97.58 | 1.48 | 97.09 | 0.14 |
| 2* X | 97.57 | 94.31 | 96.94 | 0.29 |
| 4* X | 97.24 | 96.41 | 96.05 | 0.13 |
| 6* X | 97.33 | 97.64 | 97.23 | 0.76 |
| 8* X | 97.46 | 97.85 | 96.83 | 0.45 |

## A.9 IMPACTS OF CONFIDENCE THRESHOLDS

In this section, we show the trade-off between attack success rate and accuracy when we apply the confidence threshold. We conduct an extensive evaluation to study different threshold influences on ACC and ASR. We test our framework on MNIST dataset in the continuous attack setting with three different thresholds 0.0, 0.3, and 0.7. We found that with the increase of confidence threshold, ACC is 97.2%, 96.62%, and 88.86% accordingly, in the meantime, the ASR is 22.35%, 1.93%, and 0.91% accordingly. We observe that benign local model hardening has controllable negative effects on accuracy. Meanwhile, there is a trade-off between adversarial training accuracy and standard accuracy of a model (Tsipras et al., 2019). If we aim for a much lower attack success rate, this will sacrifice part of clean accuracy. In other words, when we set a higher threshold, ASR indeed decreases, in the meantime, some low-confidence benign samples are also rejected, which causes the benign accuracy to reduce to some extent.

## A.10 DISCUSSION ON OTHER DEFENSES

In this section, we provide additional experimental results on the comparison between the Multi-KRUM (Blanchard et al., 2017) and our method. We take the CIFAR-10 dataset as an example, in the single-shot attack, Multi-KRUM can drop ASR from 80.46% to 4.18%, and our defense ASR is 7.83%. However, in the continuous attack, Multi-KRUM can only reduce ASR from 84.73% to 61.86%, and our defense ASR is 17.27%, the ACC is at a similar level. Our technique can achieve comparative performance with Multi-KRUM in the single-shot attack and outperforms Multi-KRUM in more complex attack scenarios of continuous attack. In addition, we also try to evaluate FLAME (Nguyen et al., 2021). We contacted the authors of FLAME several times for their experiments and parameters setup but got no response until submission.

## A.11 JUSTIFICATIONS FOR SOTA DEFENSES NOT WORKING

In this section, we provide concrete justifications on why SOTA defenses produce a nearly 100% attack success rate on continuous attacks setting. Continuous backdoor attacks denote that in each round the attackers will be selected and continuously participate in federated learning. We suspect there are three reasons that SOTA defenses are performing not well on continuous attacks. First, continuous backdoor attacks are more aggressive. In each round of selected participants, 40% of them are attackers and will participate in every round of model training. Second, as mentioned in (Wang et al., 2020a), even under a very low attack frequency, the attacker still manages to gradually inject the backdoor as long as federated learning runs for long enough. Third, some of their assumptions, e.g. though FoolsGold (Fung et al., 2020) assumes that benign data are non-iid, meanwhile, it also assumes manipulated data are iid, this could cause FoolsGold to be only effective under certain simpler attack scenarios, e.g. single-shot attacks.

## A.12 IMPACT OF TRIGGER QUALITY

In this section, we study the different quality of triggers effect and the evaluation results in Table 10. We evaluate CIFAR-10 dataset on single-shot attacks. We inject triggers with random shapes in both white and colorful. We observe that the randomly chosen triggers do not have much influence on attackers and cannot reduce the attack success rate. However, when we use the ground truth triggers, they do achieve the best performance in both reducing ASR on backdoor tasks and maintaining ACC on benign tasks. The evaluation shows that random triggers cannot reduce ASR and FLIP inverted

trigger can achieve comparable performance with ground truth trigger and can be further improved in the future work.

Table 10: Trigger Quality

| Single-shot Attack | CIFAR-10 | |
|---|---|---|
| | ACC | ASR |
| **No Defense** | 77.52 | 80.46 |
| **Random white trigger** | 77.89 | 81.72 |
| **Random colorful trigger** | 78.65 | 80.99 |
| **FLIP** | 73.41 | 7.83 |
| **Ground truth trigger** | 76.98 | 2.64 |

### A.13 DATA DISTRIBUTION EFFECTS

In this section, we leverage Dirichlet distribution (Minka, 2000) with a hyperparameter $\alpha$ to model different *non-i.i.d.* distributions. By increasing the hyperparameter $\alpha$ in the Dirichlet distribution, we can simulate from *non-i.i.d* to *i.i.d* distributions for the datasets (Xie et al., 2019). Here we evaluate MNIST on single-shot attacks without defense and with FLIP. We conduct an experiment consisting of different $\alpha$ from 0.2, 0.4, 0.6, 0.8, 1.0, to 2.0. The evaluation demonstrates that without defense applied, backdoor attack performance is affected by *non-i.i.d* degree. However, our defense can still cause a significant ASR degradation in different *non-i.i.d* degrees and only has slightly differences, while maintaining comparable benign classification performance.

Table 11: Data Distribution Effects

| Degree | No Defense | | FLIP | |
|---|---|---|---|---|
| | ACC | ASR | ACC | ASR |
| **0.20** | 97.72 | 75.97 | 97.56 | 0.34 |
| **0.40** | 97.47 | 80.36 | 97.64 | 0.41 |
| **0.60** | 97.01 | 80.06 | 97.64 | 0.60 |
| **0.80** | 97.13 | 81.90 | 97.49 | 1.39 |
| **1.00** | 97.80 | 81.19 | 97.53 | 1.86 |
| **2.00** | 97.82 | 93.25 | 97.31 | 1.96 |

### A.14 PROOF OF BOUNDS ON LOSS CHANGES

In this section, we will present the bound on loss changes, formulate the benign local clients training and global model aggregation process, and then provide the detailed proofs for our Theorem 1 that are related to loss changes bound. Note that, we list all the notations used in the paper in Table 12.

**Generalize Proof to complex model architectures.** Given that existing works (Xie et al., 2021; Li et al., 2021; Nguyen et al., 2021) all focus on linear models as the complexity in real-world models makes theoretical analysis tasks infeasible. Note that besides the theoretical results, empirically, we extended to non-linear and showed in Section 5 that our defense gives outstanding performance against state-of-the-art backdoor attacks, consistent with our theoretical analysis.

Throughout this paper, "clean training" refers to benign local clients training with clean data; "adversarial training" refers to benign local clients apply trigger inversion techniques to get reversed trigger, then stamp the trigger to their local clean image and assign with ground truth clean label to get the augmented dataset, then train with the augmented dataset.

In benign clients, we train with defense technique to generate trigger, then do adversarial training and submit gradients to global server. Given model parameter $W$ of one linear layer, $k$-th device holds the $n_k$ training data $\mathbf{x}_{k,n_k}$, then denoted the loss as $\mathcal{L}(W; \mathbf{x}_{k,n_k})$. Let $Y \in \{0, 1\}_i$ denote a one-hot vector of local samples. For $\mathbf{x}$, we denote $\mathbf{x}W$ as the output of the linear layer, $p_i(\mathbf{x}) = softmax(\mathbf{x}W + b)_i$ as the normalized probability for class $i$ (the output of the softmax function).

$b$ or bias is omitted in following equations for simplicity, but it would still work if added. For one example the cross-entropy loss is calculated as:

$$\mathcal{L}(x) = -\sum_i Y_i log p_i(\mathbf{x}) \tag{4}$$

$$= -\sum_i Y_i log(softmax(\mathbf{x}W)_i) \tag{5}$$

We define $G$ as the gradient for one sample:

$$G(\mathbf{x}) = \nabla l(W; \mathbf{x}, y) = \frac{d\mathcal{L}}{dw}(\mathbf{x}) = \mathbf{x}^T(p(\mathbf{x}) - Y) \tag{6}$$

Similarly, when defense technique get reversed trigger and stamp it on clean image, then we get the augmented dataset, denote is as $\mathbf{x}_{aug}$, then the gradient on augmented dataset $G'$ can be written as:

$$G'(\mathbf{x}_{aug}) = \nabla l(W; \mathbf{x}_{aug}, y) = \mathbf{x}_{aug}^T(p(\mathbf{x}_{aug}) - Y)) \tag{7}$$

Here, we describe one around (say the $r$-th) of the standard $FedAvg$ algorithm. When the benign device in $k$-th receive the global weights $W_r$, and then performs $E (= 1)$ local updates (lets $W_r^k = W_r$), in benign clients, we training on both clean dataset and augmented dataset:

$$
\begin{aligned}
W_{r+1}^k &\leftarrow W_r^k - \eta_r \nabla F_k(W_r^k, \xi_r^k) \\
&\leftarrow W_r^k - \eta_r [\sum_{j=1}^{n_k} [\mathbf{x}_j^{T,k}(p(\mathbf{x}_j^k) - Y_j)] + \sum_{j=1}^{n_k} [\mathbf{x}_{aug,j}^{T,k}(p(\mathbf{x}_{aug,j}^k) - Y_j)]] \\
&\leftarrow W_r^k - \eta_r \sum_{j=1}^{n_k} [\mathbf{x}_j^{T,k}(p(\mathbf{x}_j^k) - Y_j)] - \eta_r \sum_{j=1}^{n_k} [\mathbf{x}_{aug,j}^{T,k}(p(\mathbf{x}_{aug,j}^k) - Y_j)]
\end{aligned} \tag{8}
$$

where $\eta_r$ is the learning rate (a.k.a. step size), $n_k$ is the number of samples in $k$-th client.

In global server, define $\delta$ as the malicious clients generated trigger, $\delta + \epsilon$ as the benign clients generated trigger, then we can represent backdoored sample as $(\mathbf{x} + \delta)$ and augmented sample as $(\mathbf{x} + \delta + \epsilon)$. Benign clients updates can be written as:

$$W_{r+1}^k \leftarrow W_r^k - \eta_r \sum_{j=1}^{n_k} [\mathbf{x}_j^{T,k}(p(\mathbf{x})_j^k) - Y_j^k)] - \eta_r \sum_{j=1}^{n_k} [(\mathbf{x}_j + \delta + \epsilon)^{T,k}(p(\mathbf{x}_j + \delta + \epsilon)^k) - Y_j^k)] \tag{9}$$

In the threat model, we consider the practical oblivious but honest attack setting that a defender has no control on malicious clients and they can perform any kinds of attack, as long as attackers follow the federated learning protocol. Our proof focuses on the two-client setting, one benign and the other malicious. Thus, we represent the malicious clients updates as $W_M$.

After each local finished their training, they submit their model updates to global. Then global aggregation step performs

$$W_{r+1} \leftarrow \eta_r \sum_{k=1}^{N} g_k W_{r+1}^k \tag{10}$$

$g_k$ is the weight of the $k$-th device. In order to simplify, here we take $g_k$ as 1 and assume we only have two clients (N=2), $k = 1$ is benign client, $n_1$ denotes the number of samples in this benign client. Then the aggregated global weight are the each local weights aggregate together. Then the

aggregated global weight can be written as

$$
\begin{aligned}
W_{r+1} &= \sum_{k=1}^{N} W_{r+1}^{k} \\
&= W_{r+1}^{1} + W_{r+1}^{2} \\
&= -\eta_r \sum_{j=1}^{n_1} [\mathbf{x}_j^{T,k}(p(x)_j^k) - Y_j^k)] - \eta_r \sum_{j=1}^{n_1} [(\mathbf{x}_j + \delta + \epsilon)^{T,1}(p(\mathbf{x}_j + \delta + \epsilon)^1 - Y_j^1)] \\
&\quad + 2W_r - W_M \\
&= -\eta_r \sum_{j=1}^{n_1} [\mathbf{x}_j^{T,k}(p(x)_j^k) - Y_j^k)] - \eta_r \sum_{j=1}^{n_1} [(\mathbf{x}_j + \delta + \epsilon)^{T,1}(p(\mathbf{x}_j + \delta + \epsilon)^1 - Y_j^1)] \\
&\quad + 2W_r - W_M
\end{aligned}
\tag{11}
$$

When we consider the without defense setting, $\delta + \epsilon$ not exists, in round $t + 1$, $W_{t+1}$ the global weight without local weights can be written as

$$
W_{r+1} = -\eta_r \sum_{j=1}^{n_1} [\mathbf{x}_j^{T,1}(p(x)_j^1 - Y_j^1)] + 2W_r - W_M
\tag{12}
$$

When we consider the with defense setting, $\delta + \epsilon$ exists, in round $t + 1$, $W_{t+1}$ the global weight without local weights can be written as

$$
\begin{aligned}
W'_{r+1} &= -\eta_r \sum_{j=1}^{n_1} [\mathbf{x}_j^{T,1}(p(x)_j^1 - Y_j^1)] - \eta_r \sum_{j=1}^{n_1} [(\mathbf{x}_j + \delta + \epsilon)^{T,1}(p(\mathbf{x}_j + \delta + \epsilon)^1 - Y_j^1)] \\
&\quad + 2W_r - W_M
\end{aligned}
\tag{13}
$$

The difference between with defense and without defense training is exactly how much adversarial training in benign will influence other clients, it can be written as

$$
W'_{r+1} - W_{r+1} = -\eta_r \sum_{j=1}^{n_1} [(\mathbf{x}_j + \delta + \epsilon)^{T,1}(p(\mathbf{x}_j + \delta + \epsilon)^1 - Y_j^1)]
\tag{14}
$$

Given model parameter $W$ of one linear layer, the $k$-th device holds $n_k$ training data $\{\mathbf{x}_{k,j}, y_{k,j}\}_{j=1}^{n_k}$. We denote the loss as $\mathcal{L}(W; \{\mathbf{x}_{k,j}, y_{k,j}\}_{j=1}^{n_k})$. Denote $\mathbf{x}W$ as the output of the linear layer, $P_i(x) = softmax(\mathbf{x}W + b)_i$ as the normalized probability for class $i$ (the output of the *softmax* function). We omit $b$ (bias) in the following theoretical analysis for simplicity. Adding the bias term to our analysis is straightforward.

Global softmax cross-entropy loss function can be written as:

$$
\begin{aligned}
\mathcal{L}_{global} &= -\sum_{i=1}^{I} q_i log(p_i) \\
&= -\sum_{i=1}^{I} q_i log softmax(\mathbf{x}W)_i \\
&= -\sum_{i=1}^{I} q_i log\left(\frac{e^{(\mathbf{x}W)_i}}{\sum_{t=1}^{I} e^{(\mathbf{x}W)_t}}\right) \\
&= -\sum_{i=1}^{I} q_i(\mathbf{x}W)_i + log\left(\sum_{t=1}^{I} e^{(\mathbf{x}W)_t}\right) \\
&= -\sum_{i=1}^{I} q_i(\mathbf{x}W)_i + log\left(\sum_{t=1}^{I} e^{(\mathbf{x}W)_t}\right)
\end{aligned}
\tag{15}
$$

Since we want to compare the loss changes in two different cases (e.g. with defense and without defense setting), to observe if the dedution of the loss increase or decrease, here we let the two losses (say $\mathcal{L}'_g$ is with defense, $\mathcal{L}_g$ is without defense) deduct each other:

$$
\begin{aligned}
\mathcal{L}'_g - \mathcal{L}_g &= -\sum_{i=1}^{I} q_i(\mathbf{x}W')_i + log(\sum_{t=1}^{I} e^{(\mathbf{x}W')_t}) + \sum_{i=1}^{I} q_i(\mathbf{x}W)_i - log(\sum_{t=1}^{I} e^{(\mathbf{x}W)_t}) \\
&= -\sum_{i=1}^{I} q_i[(W'-W)\mathbf{x}]_i + log(\sum_{t=1}^{I} e^{(\mathbf{x}W')_t}) - log(\sum_{t=1}^{I} e^{(\mathbf{x}W)_t}) \qquad (16) \\
&= -\sum_{i=1}^{I} q_i[(W'-W)\mathbf{x}]_i + log(\frac{\sum_{t=1}^{I} e^{(\mathbf{x}W')_t}}{\sum_{t=1}^{I} e^{(\mathbf{x}W)_t}})
\end{aligned}
$$

Since both $\sum_{t=1}^{I} e^{(\mathbf{x}W')_t}$ and $\sum_{t=1}^{I} e^{(\mathbf{x}W)_t}$ are a sequence of positive numbers. Then from Mitrinović & Vasić (1970) we can have an inequality of

$$
\min_{1\le t\le I} \frac{e^{(\mathbf{x}W')_t}}{e^{(\mathbf{x}W)_t}} \le \frac{\sum_{t=1}^{I} e^{(\mathbf{x}W')_t}}{\sum_{t=1}^{I} e^{(\mathbf{x}W)_t}} \le \max_{1\le t\le I} \frac{e^{(\mathbf{x}W')_t}}{e^{(\mathbf{x}W)_t}} \qquad (17)
$$

*Proof.* If we denote $m = \min_t \frac{e^{(\mathbf{x}W')_t}}{e^{(\mathbf{x}W)_t}}$ and $M = \max_t \frac{e^{(\mathbf{x}W')_t}}{e^{(\mathbf{x}W)_t}}$, then we have successively

$$
m \le \frac{e^{(\mathbf{x}W')_t}}{e^{(\mathbf{x}W)_t}} \le M \qquad (18)
$$

$$
m \cdot e^{(\mathbf{x}W)_t} \le e^{(\mathbf{x}W')_t} \le M \cdot e^{(\mathbf{x}W)_t} \qquad (19)
$$

$$
m \cdot \sum_{t=1}^{I} e^{(\mathbf{x}W)_t} \le \sum_{t=1}^{I} e^{(\mathbf{x}W')_t} \le M \cdot \sum_{t=1}^{I} e^{(\mathbf{x}W)_t} \qquad (20)
$$

$$
m \le \frac{\sum_{t=1}^{I} e^{(\mathbf{x}W')_t}}{\sum_{t=1}^{I} e^{(\mathbf{x}W)_t}} \le M \qquad (21)
$$

$\square$

Then we can get

$$
\min_{1\le t\le I} \frac{e^{(\mathbf{x}W')_t}}{e^{(\mathbf{x}W)_t}} \le \frac{\sum_{t=1}^{I} e^{(\mathbf{x}W')_t}}{\sum_{t=1}^{I} e^{(\mathbf{x}W)_t}} \le \max_{1\le t\le I} \frac{e^{(\mathbf{x}W')_t}}{e^{(\mathbf{x}W)_t}} \qquad (22)
$$

By using log function monotonicity property, we can get an inequality of

$$
log m \le log(\frac{\sum_{t=1}^{I} e^{(\mathbf{x}W')_t}}{\sum_{t=1}^{I} e^{(\mathbf{x}W)_t}}) \le log M \qquad (23)
$$

So the deduction of $\mathcal{L}'_g - \mathcal{L}_g$ can be written as:

$$
log m - \sum_{i=1}^{I} q_i[\mathbf{x}(W'-W)]_i \le \mathcal{L}'_g - \mathcal{L}_g \le log M - \sum_{i=1}^{I} q_i[\mathbf{x}(W'-W)]_i \qquad (24)
$$

$$
log \min_t \frac{e^{(\mathbf{x}W')_t}}{e^{(\mathbf{x}W)_t}} - \sum_{i=1}^{I} q_i[\mathbf{x}(W'-W)]_i \le \mathcal{L}'_g - \mathcal{L}_g \le log \max_t \frac{e^{(\mathbf{x}W')_t}}{e^{(\mathbf{x}W)_t}} - \sum_{i=1}^{I} q_i[\mathbf{x}(W'-W)]_i
$$
$$
(25)
$$

Denote the left hand side of above formula as $\Delta \min \_loss$, denote the inequality's right hand side value as $\Delta \max \_loss$.

$$
\begin{aligned}
\Delta \min \_loss &= log \min_t \frac{e^{(\mathbf{x}W')_t}}{e^{(\mathbf{x}W)_t}} - \sum_{i=1}^{I} q_i [\mathbf{x}(W' - W)]_i \\
&= \min_t log \frac{e^{(\mathbf{x}W')_t}}{e^{(\mathbf{x}W)_t}} - \sum_{i=1}^{I} q_i [\mathbf{x}(W' - W)]_i \\
&= \min_t log \, e^{[\mathbf{x}(W' - W)]_t} - \sum_{i=1}^{I} q_i [\mathbf{x}(W' - W)]_i \\
&= \min_t [\mathbf{x}(W' - W)]_t - \sum_{i=1}^{I} q_i [\mathbf{x}(W' - W)]_i
\end{aligned}
\tag{26}
$$

Then we can get the lower bound and upper bound of $\mathcal{L}'_g - \mathcal{L}_g$

$$
\min_t [\mathbf{x}(W' - W)]_t - \sum_{i=1}^{I} q_i [\mathbf{x}(W' - W)]_i \leq \mathcal{L}'_g - \mathcal{L}_g \leq \max_t [\mathbf{x}(W' - W)]_t - \sum_{i=1}^{I} q_i [\mathbf{x}(W' - W)]_i
\tag{27}
$$

Let $\mathcal{L}'_g$ denote the global model loss with defense, $\mathcal{L}_g$ as without defense, let $\Delta W = W' - W$ denote the weight differences with and without defense. The loss difference with and without defense can be upper and lower bounded by (as shown in Theorem 1)

$$
\min_t (\mathbf{x}\Delta W)_t - \sum_{i=1}^{I} q_i (\mathbf{x}\Delta W)_i \leq \mathcal{L}'_g - \mathcal{L}_g \leq \max_t (\mathbf{x}\Delta W)_t - \sum_{i=1}^{I} q_i (\mathbf{x}\Delta W)_i
\tag{28}
$$

To facilitate the analysis, we denote the upper bound as $\Delta \max \_loss$ and the lower bound as $\Delta \min \_loss$. To efficiently reduce the attack success rate and maintain the clean accuracy, we studied this lower bound on backdoor data, which indicates the minimal improvements on the backdoor defense. Similarly we studied the upper bound for clean data, as they indicates the worst-case accuracy degradation.

Denote the number of backdoor samples as $n_b$ and the number of benign samples as $n_c$. Note backdoor samples are written as $\mathbf{x}_s + \delta$. By using Theorem 1, we have $\Delta \min \_loss = \sum_{s=1}^{n_b} \min_t [(\mathbf{x}_s + \delta)\Delta W]_t - \sum_{s=1}^{n_b} \sum_{i=1}^{I} q_{s,i}[(\mathbf{x}_s + \delta)\Delta W]_i$. And similarly on benign data, we have $\Delta \max \_loss$ $\mathbf{x}_s$, $\Delta \max \_loss = \sum_{s=1}^{n_c} \max_t (\mathbf{x}_s \Delta W)_t - \sum_{s=1}^{n_c} \sum_{i=1}^{I} q_{s,i}(\mathbf{x}_s \Delta W)_i$.

### A.15 PROOF OF GENERAL ROBUSTNESS CONDITION

In this section, we will present general condition of robustness on trigger generation, formulate $\Delta min\_loss$ on backdoored data and $\Delta max\_loss$ on clean data, and then provide the detailed proofs for our Theorem 2 that are related to general robustness condition.

Our intuition is that we want the loss to increase more on backdoored data, and increase less on clean data. This means after applying defense, the global server loss in backdoored data will increase and the loss in clean data will change within a constant range. Accordingly, when evaluate on $n_b$ backdoored data, we want the lower bound at least greater than 0, $\Delta min\_loss \geq 0$. When evaluate on $n_c$ clean data, we want the upper bound $\Delta max\_loss \leq \zeta$, here $\zeta$ is a constant. In evaluation, denote global server has $n_b$ backdoored data and $n_c$ clean data for testing.

When evaluating on $n_b$ backdoored data

$$
\begin{aligned}
\mathcal{L}'_g - \mathcal{L}_g &\geq \sum_{s=1}^{n_b} \min_t [(\mathbf{x}_s + \delta)(W' - W)]_t - \sum_{s=1}^{n_b} \sum_{i=1}^{I} q_{s,i}[(\mathbf{x}_s + \delta)(W' - W)]_i \\
&= \Delta \min \_loss \geq 0
\end{aligned}
\tag{29}
$$

When evaluating on $n_c$ clean data

$$
\mathcal{L}'_g - \mathcal{L}_g \leq \sum_{s=1}^{n_c} \max_t [\mathbf{x}_s(W' - W)]_t - \sum_{s=1}^{n_c} \sum_{i=1}^{I} q_{s,i}[\mathbf{x}_s(W' - W)]_i \tag{30}
$$
$$
= \Delta \max\_loss \leq \zeta
$$

Since previous results we know $W'_{r+1} - W_{r+1}$ can be represented as

$$
W'_{r+1} - W_{r+1} = -\eta_r \sum_{j=1}^{n_1} [(\mathbf{x}_j + \delta + \epsilon)^T (p(\mathbf{x}_j + \delta + \epsilon) - Y_j)] \tag{31}
$$
$$
= \eta_r \sum_{j=1}^{n_1} [(\mathbf{x}_j + \delta + \epsilon)^T (Y_j - p(\mathbf{x}_j + \delta + \epsilon))]
$$

We denote $q_s^*$ as a one-hot vector for sample $s$ with $I$ dimensions. Its $i$-th dimension is defined as $q_{s,i}^*$, and $q_{s,i}^* = 1$ if $i = \arg\min_t [(\mathbf{x}_s + \delta)(W' - W)]_t$.

$$
\Delta \min\_loss = \sum_{s=1}^{n_b} \min_t [(\mathbf{x}_s + \delta)(W' - W)]_t - \sum_{s=1}^{n_b} \sum_{i=1}^{I} q_{s,i}[(\mathbf{x}_s + \delta)(W' - W)]_i
$$
$$
= \eta_r \sum_{s=1}^{n_b} \sum_{i=1}^{I} q_{s,i}^* [(\mathbf{x}_s + \delta) \sum_{j=1}^{n_1} [(\mathbf{x}_j + \delta + \epsilon)^T (Y_j - p(\mathbf{x}_j + \delta + \epsilon))]]_i
$$
$$
- \eta_r \sum_{s=1}^{n_b} \sum_{i=1}^{I} q_{s,i} [(\mathbf{x}_s + \delta) \sum_{j=1}^{n_1} [(\mathbf{x}_j + \delta + \epsilon)^T (Y_j - p(\mathbf{x}_j + \delta + \epsilon))]]_i
$$
$$
= \eta_r \sum_{s=1}^{n_b} \sum_{i=1}^{I} (q_{s,i}^* - q_{s,i}) [(\mathbf{x}_s + \delta) \eta_r \sum_{j=1}^{n_1} [(\mathbf{x}_j + \delta + \epsilon)^T (Y_j - p(\mathbf{x}_j + \delta + \epsilon))]]_i \tag{32}
$$

Let $z_s = \mathbf{x}_s + \delta + \epsilon$ and $z_j = \mathbf{x}_j + \delta + \epsilon$, then the $\Delta \min\_loss$ is

$$
\Delta \min\_loss = \eta_r \sum_{s=1}^{n_b} \sum_{i=1}^{I} (q_{s,i}^* - q_{s,i}) [(z_s - \epsilon) \sum_{j=1}^{n_1} [z_j^T (Y_j - p(z_j))]]_i \tag{33}
$$

Let

$$
f(\epsilon) = \Delta \min\_loss
$$
$$
= \eta_r \sum_{s=1}^{n_b} \sum_{i=1}^{I} (q_{s,i}^* - q_{s,i}) \{(z_s - \epsilon) \sum_{j=1}^{n_1} [z_j^T (Y_j - p(z_j))]\}_i \tag{34}
$$

Compute the gradient of $f(\epsilon)$, we have

$$
\frac{\nabla f(\epsilon)}{\nabla \epsilon} = -\eta_r \sum_{s=1}^{n_b} \sum_{i=1}^{I} (q_{s,i}^* - q_{s,i}) [\sum_{j=1}^{n_1} [z_j^T (Y_j - p(z_j))]]_i \tag{35}
$$

Let $||\epsilon||_\infty \leq \alpha$, and that $f(\epsilon)$ is a linear function, we know the minimal value of $f(\epsilon)$ is achieved when

$$
\epsilon_k = \alpha \, \text{sign} \left\{ \eta_r \sum_{s=1}^{n_b} \sum_{i=1}^{I} (q_{s,i}^* - q_{s,i}) \sum_{j=1}^{n_1} [z_j^T (Y_j - p(z_j))]]_{i,k} \right\} \tag{36}
$$

For simplicity, denote $b_k$ as

$$
b_k = \text{sign} \left\{ \eta_r \sum_{s=1}^{n_b} \sum_{i=1}^{I} (q_{s,i}^* - q_{s,i}) \sum_{j=1}^{n_1} [z_j^T (Y_j - p(z_j))]]_{i,k} \right\} \tag{37}
$$

and the vector as $\mathbf{b} = [b_1, ..., b_d]$. The minimal condition is thus $\epsilon = \alpha \mathbf{b}$.

Replace $\epsilon = \alpha \mathbf{b}$ into eq. (34), and in order to be consistent with previous section in main text Theorem 2, we use $\mathbf{q_j}$ to replace $Y_j$, we have

$$f(\epsilon) \geq \eta_r \sum_{s=1}^{n_b} \sum_{i=1}^{I} (q_{s,i}^* - q_{s,i})\{(\mathbf{z_s} - \alpha \mathbf{b}) \sum_{j=1}^{n_1} [\mathbf{z_j}^T (\mathbf{q_j} - p(\mathbf{z_j}))]\}_i \tag{38}$$

The sufficient condition of $f(\epsilon) \geq 0$ is thus

$$\eta_r \sum_{s=1}^{n_b} \sum_{i=1}^{I} (q_{s,i}^* - q_{s,i})\{(\mathbf{z_s} - \alpha \mathbf{b}) \sum_{j=1}^{n_1} [\mathbf{z_j}^T (\mathbf{q_j} - p(\mathbf{z_j}))]\}_i \geq 0 \tag{39}$$

$$\alpha \mathbf{b}\{\eta_r \sum_{s=1}^{n_b} \sum_{i=1}^{I} (q_{s,i}^* - q_{s,i})\{\sum_{j=1}^{n_1} [\mathbf{z_j}^T (\mathbf{q_j} - p(\mathbf{z_j}))]\}_i\}$$
$$\leq \eta_r \sum_{s=1}^{n_b} \sum_{i=1}^{I} (q_{s,i}^* - q_{s,i})\{\mathbf{z_s} \sum_{j=1}^{n_1} [\mathbf{z_j}^T (\mathbf{q_j} - p(\mathbf{z_j}))]\}_i \tag{40}$$

Note that for any vector $\mathbf{x}$, we have $\text{sign}(\mathbf{x})\mathbf{x} \geq 0$. And we can divide the right hand side by the left hand side and finish the prove.

$$\alpha \leq \frac{\eta_r \sum_{s=1}^{n_b} \sum_{i=1}^{I} (q_{s,i}^* - q_{s,i})\{\mathbf{z_s} \sum_{j=1}^{n_1} [\mathbf{z_j}^T (\mathbf{q_j} - p(\mathbf{z_j}))]\}_i}{\mathbf{b}\{\eta_r \sum_{s=1}^{n_b} \sum_{i=1}^{I} (q_{s,i}^* - q_{s,i})\{\sum_{j=1}^{n_1} [\mathbf{z_j}^T (\mathbf{q_j} - p(\mathbf{z_j}))]\}_i\}} \tag{41}$$

Each term in above can be computed, then we can always find a small enough error range $\epsilon$ where surely improve the loss function.

Similarly, in upper bound of $\Delta \max\_loss$, we denote $q_s^*$ as a one-hot vector, denote $q_s^*$ as a one-hot vector for sample $s$ with $I$ dimensions. Its $i$-th dimension is defined as $q_{s,i}^*$, and $q_{s,i}^* = 1$ if $i = argmax_t[\mathbf{x}_s(W' - W)]_t$, let

$$g(\epsilon) = \Delta \max\_loss$$
$$= \eta_r \sum_{s=1}^{n_c} \sum_{i=1}^{I} (q_{s,i}^* - q_{s,i})\mathbf{x_s} \sum_{j=1}^{n_1} [\mathbf{z_j}^T (\mathbf{q_j} - p(\mathbf{z_j}))]_i \tag{42}$$

Note that $g(\epsilon)$ is nothing but a constant with respect to $\epsilon$. This means that the upper bound loss is up to some constant with respect to the recovered trigger $z_j$.

Note that $\Delta \min\_loss \geq 0$ indicates that the defense is provably effective than without defense. Since benign local clients training can increase global model backdoor loss, and they have positive effects on mitigating malicious poisoning effect. The second condition $\Delta \max\_loss \leq \eta_r \sum_{s=1}^{n_c} \sum_{i=1}^{I} (q_{s,i}^* - q_{s,i})\mathbf{x_s} \sum_{j=1}^{n_1} [\mathbf{z_j}^T (\mathbf{q_j} - p(\mathbf{z_j}))]_i$ indicates that the defense is provably guarantee maintaining similar accuracy on clean data.

## A.16 GLOSSARY OF NOTATIONS

Table 12: Glossary of notations

| Notation | Description |
|---|---|
| $\mathbf{x}_{k,j}, y_{k,j}$ | the $k$-th client device $j$-th data sample and its label |
| $q_{s,i}$ | $s$-th sample $i$-th dimension |
| $\eta_r$ | the learning rate (a.k.a. step size) |
| $W_r^k$ | the $k$-th client device in $r$-th round weights |
| $W$ | local model weights *without* defense |
| $W'$ | local model weights *with* defense |
| $\tau$ | confidence threshold |
| $R_b$ | the number of rejected backdoor samples *without* defense |
| $R_b'$ | the number of rejected backdoor samples *with* defense |
| $R_{bd} = R_b' - R_b$ | the number of backdoored samples that are rejected after defense applied |
| $R_c$ | the number of rejected benign samples *without* defense |
| $R_c'$ | the number of rejected benign samples *with* defense |
| $R_{bn} = R_c' - R_c$ | the number of benign samples that are rejected after defense applied |
| $\delta$ | the ground truth trigger |
| $\delta + \epsilon$ | the various trigger inversion technique recovered trigger |
| $\epsilon$ | the difference between the reversed trigger and the ground truth trigger |
| $\mathbf{z} = \mathbf{x} + \delta + \epsilon$ | $\mathbf{z}$ denotes the benign sample stamped with the recovered trigger |
| $\mathcal{L}_g$ | the global model loss *without* defense |
| $\mathcal{L}_g'$ | the global model loss *with* defense |

