# OpenReview forum: "FLIP: A Provable Defense Framework for Backdoor Mitigation in Federated Learning"
_ICLR.cc/2023/Conference — ICLR 2023 poster_

### Official Review · Reviewer_zr2z · 2022-10-20

**Confidence:** 5
**Correctness:** 3
**Technical Novelty And Significance:** 2
**Empirical Novelty And Significance:** Not applicable
**Recommendation:** 3

**Clarity, Quality, Novelty And Reproducibility:**





=====Theoretical results====

My main concern is about the two theoretical results, i.e., Theorem 1 and Theorem 2, that are inaccurate.

In Theorem 1, the loss Lg’ and Lg defined is unclear, so does x. When I went through the appendix, I realized  Lg’ and Lg are defined on a single data point x, instead of a training set.
However, when generalizing it to the set of samples, the relationship considered in Eqn (3) is inaccurate.
More specifically, the accurate form in Eqn (3) should sum_{s=1}^n_b  min_t (x \nabla W),  instead of min_t (sum_{s=1}^n_b  x \nabla W), as these two terms are not equal in general.
Therefore, the results in theorem 2 based on Eqn(3) is inaccurate.

Moreover, the theoretical results are under a very simple linear model. A key reason making backdoor attacks be successful is due to the overfitting/memorization of deep models on the trigger.  Hence, I am doubtful that these theoretical results can guide the defense effectiveness.

====Evaluations====
What are the recovered triggers by the proposed defense? Neural Cleanse often only recovers very blurred trigger. What if benign clients just inject a set of randomly chosen triggers for adversarial training?  What is this defense performance? Further, is the best performance achieved when the true trigger is used for adversarial training?  Basically, I would like to see how the  trigger affects the performance.

How many extra augmented data samples are used?

Non-IID is an important factor that affects the defense performance. What’s the defense performance against more non-IID data across clients?

In adaptive attacks, is the setting that the malicious clients set the inverted trigger + clean images as a targeted label, while benign clients set the inverted trigger + clean images as the true label? If this is correct, what’s the insight that makes the defense effective? More specifically, if malicious clients have more poisoned samples than benign clients’ clean samples, and the deep model memorizes the trigger, how could it be possible that the defense is effective?

The proposed defense does not compare with tee state-of-the-art, e.g.,

Shejwalkar and Houmansadr, Manipulating the Byzantine: Optimizing Model Poisoning Attacks and Defenses for Federated
Learning, In NDSS, 2021

How the class distance is defined? Just using the data samples from the source-target class? How the number of samples affect this distance?


**Strength And Weaknesses:**

Strengths
+ The studied problem is important
+ The paper proposes a three stage defense against backdoor attacks to federated learning


Weaknesses
-Theoretical results are inaccurate
-Theoretical results do not provide meaningful guidance about why the propose defense works
-Evaluation is not sufficient




**Summary Of The Paper:**

The paper proposes a defense against backdoor attacks to federated learning. Existing defenses do not study how hardening benign clients can affect the global model (and the malicious clients).  The paper proposes a three stage defense that aims to harden benign clients and connect cross-entropy loss, attack success rate, and clean accuracy.  The proposed defense is evaluated on three datasets and shows better performance than previous defenses.




**Summary Of The Review:**

The studied problem is important. The proposed defense is somewhat novelty. However, the proposed defense is not well justified, and the theoretical results may be inaccurate and cannot be generalized to general deep nonlinear models.

---

> ### Author Response · Authors · 2022-11-17
> **Official Response to Reviewer zr2z (1/3)**
>
> ====Theoretical results====
> > 1. The relationship considered in Eqn (3) is inaccurate. More specifically, the accurate form in Eqn (3) should sum_{s=1}^n_b min_t (x \nabla W), instead of min_t (sum_{s=1}^n_b x \nabla W), as these two terms are not equal in general.
>
> We thank the reviewer for the insightful and valuable comment. The reviewer was right. We meant sum_{s=1}^n_b min_t (x \nabla W) while our submission mistakenly used min_t (sum_{s=1}^n_b x \nabla W). However, the typo did not affect the overall validity of our proofs. We thoroughly checked our proofs after fixing the mistake to ensure their correctness. We have highlighted the changes in the proof and updated the corresponding theoretical statements in blue text.
>
>
> > 2. The theoretical results are under a very simple linear model. I am doubtful that these theoretical results can guide the defense effectiveness.
>
> We thank the reviewer for the insightful question. We note that existing works on provable defenses for backdoor attacks in FL [3, 4, 5, 6] also focused on linear models as the complexity in real-world models makes theoretical analysis tasks infeasible. Note that besides the theoretical results, empirically, we extended to non-linear and showed in Section 5 (Experiment) that our defense gives outstanding performance against state-of-the-art backdoor attacks, consistent with our theoretical analysis.
>
> We have the discussion in Appendix A.12 (Generalize Proof to complex model architectures) and will make the discussion clear in our final version.
>
> ====Evaluations====
> > 3. What are the recovered triggers by the proposed defense? What if benign clients just inject a set of randomly chosen triggers for adversarial training? What is this defense performance? Is the best performance achieved when the true trigger is used for adversarial training? Basically, I would like to see how the trigger affects performance.
>
> The triggers recovered by FLIP did not look exactly like the ground truth triggers but they shared common features. Examples of ground truth triggers and inverted triggers can be found in Appendix A.1. There was certain randomness in FLIP’s trigger recovery (e.g., by random initialization). We did an experiment during rebuttal of running trigger recovery with different initial values and used them in hardening. The results (i.e., ASR and ACC) hardly changed. In addition, we injected triggers with random shapes and colors on benign clients, and they did not improve model robustness as these triggers did not share common features with the ground truth triggers. Our results were shown below and in Appendix A.15 Table 11. Observe that FLIP inverted triggers can achieve comparable performance as those with ground truth triggers.
>
>
>
> | **Single-shot Attack**      | **CIFAR10** |       |
> |-----------------------------|-------------|-------|
> |                             | ACC         | ASR   |
> | **No Defense**              | 77.52       | 80.46 |
> | **Random white trigger**    | 77.89       | 81.72 |
> | **Random colorful trigger** | 78.65       | 80.99 |
> | **FLIP**                    | 73.41       | 7.83  |
> | **Ground truth trigger**    | 76.98       | 2.64  |
>
>
>
> > 4. How many extra augmented data samples are used?
>
> We used 64 augmented samples in each batch of each local client training, following the same setting in [7]. We included this in Appendix A.1.

---

> > ### Author Response · Authors · 2022-11-17
> > **Official Response to Reviewer zr2z (2/3)**
> >
> > > 5. Non-IID is an important factor that affects the defense performance. What’s the defense performance against more non-IID data across clients?
> >
> > We thank the reviewer for the insightful question. During rebuttal, we studied the different non-iid factor effects and updated the evaluation in the revision. We leveraged Dirichlet distribution [8] with a hyperparameter $\alpha$ to model different \textit{non-i.i.d.} distributions.
> >
> > By increasing the hyperparameter $\alpha$ in the Dirichlet distribution, we could simulate from \textit{non-i.i.d} to \textit{i.i.d} distributions for the datasets [9].
> > Here we evaluated MNIST on single-shot attacks without defense and with FLIP. We conducted an experiment consisting of different $\alpha$ from 0.2, 0.4, 0.6, 0.8, 1.0, to 2.0. The evaluation demonstrated that without defense applied, the backdoor attack performance was affected by the \textit{non-i.i.d} degree. However, our defense could still cause a significant ASR degradation in different \textit{non-i.i.d} degrees and only had slightly differences, while maintaining comparable benign classification performance.
> >
> > We included the discussion and evaluation in Appendix A.16 Table 12 in our revision.
> >
> > |            | **No Defense** |       | **FLIP** |      |
> > |------------|:----------------:|:-------:|----------|------|
> > | **Degree** | ACC            | ASR   | ACC      | ASR  |
> > | **0.20**   | 97.72          | 75.97 | 97.56    | 0.34 |
> > | **0.40**   | 97.47          | 80.36 | 97.64    | 0.41 |
> > | **0.60**   | 97.01          | 80.06 | 97.64    | 0.60 |
> > | **0.80**   | 97.13          | 81.90 | 97.49    | 1.39 |
> > | **1.00**   | 97.80          | 81.19 | 97.53    | 1.86 |
> > | **2.00**   | 97.82          | 93.25 | 97.31    | 1.96 |
> >
> >
> >
> >
> >
> > > 6. More specifically, if malicious clients have more poisoned samples than benign clients’ clean samples, and the deep model memorizes the trigger, how could it be possible that the defense is effective?
> >
> > We thank the reviewer for bringing up the interesting point. We discussed this point in two parts, the first is the federated learning attackers' assumption, and the second is the effects of injecting a large number of poison samples.
> >
> > Firstly, existing work [2] mentioned that the number of attackers is usually around 25% - 50%. In a Byzantine attack setting [10], attackers are usually less than one-third of the total participants. Even in the most aggressive FL backdoor attack setting [9], there are 40% attackers. If we take the trigger as a feature that the model can learn, then the benign client and the malicious client are actually learning the same trigger feature but they are labeling the trigger feature differently. So the prediction confidence of the trigger feature will also be very low, as long as the reversed trigger quality satisfies our given bound.
> >
> > Secondly, existing work [9] has an observation, which is, if the malicious users apply a lot more poisoned samples or scale up the weights, this indeed will lead to more influential and resistant attack performance, however, this will also cause the failure of global model in the main benign task [9].
> >
> >
> > > 7. The proposed defense does not compare with tee state-of-the-art, e.g.,
> > Shejwalkar and Houmansadr, Manipulating the Byzantine: Optimizing Model Poisoning Attacks and Defenses for Federated Learning, In NDSS, 2021
> >
> > We thank the reviewer for pointing out the interesting related work of Shejwalkar, Virat, et al [1]. The authors open-sourced their attack code, but not the defense code. We contacted the authors during the rebuttal for their implementations and received no response. As an alternative, we implemented it by ourselves based on their paper. We provided a comparative experiment between our FLIP framework with DnC, and we updated our related work discussion with both [1] and [2], and also the experimental results in Table 1 and Table 2 in our revision paper.
> >
> > From the experiment, we can see, DnC performs well in single-shot attack settings. However, in more aggressive and complex continuous attack settings, DnC cannot remove the backdoor and will even decrease the accuracy. However, FLIP can reduce ASR and maintain benign classification performance across different settings.
> >
> > |  **Single-shot attack**  | **DnC** |      | **FLIP** |      |
> > |:-----------------:|:-------:|:----:|:--------:|:----:|
> > |                   |   ACC   |  ASR |    ACC   |  ASR |
> > | **MNIST**         | 97.61   | 0.46 | 96.05    | 0.13 |
> > | **Fashion-MNIST** | 80.33   | 5.35 | 78.20    | 3.16 |
> > | **CIFAR-10**      | 76.66   | 5.36 | 73.41    | 7.83 |
> >
> > |   **Continuous attack**  | **DnC** |       | **FLIP** |       |
> > |:-----------------:|:-------:|:-----:|:--------:|:-----:|
> > |                   |   ACC   |  ASR  |    ACC   |  ASR  |
> > |     **MNIST**     | 89.81   | 99.78 | 96.62    | 1.93  |
> > | **Fashion-MNIST** | 65.90   | 97.73 | 72.99    | 17.65 |
> > |    **CIFAR-10**   | 53.16   | 82.37 | 71.28    | 22.90 |

---

> > > ### Author Response · Authors · 2022-11-17
> > > **Official Response to Reviewer zr2z (3/3)**
> > >
> > > > 8. How the class distance is defined? Just using the data samples from the source-target class? How the number of samples affect this distance?
> > >
> > > The class distance is defined by the trigger size, we use $L^{1}$ norm to measure. The intuition here is that if it’s easy to generate a small trigger from the source class to the target class,  the distance between the two classes is small. Otherwise, the class distance is large. Furthermore, the model is robust when all the class distances are large, otherwise one can easily generate a small trigger between the two classes.
> > >
> > > The class distances of models in the wild are very small and do not align well with humans' intuition, for example, classes turtle and bird have smaller distances than classes cat and dog, which makes the models vulnerable to backdoor attacks [7]. The class distance is independent of the number of local client samples, in other words, the class distance is related to the model’s robustness itself instead of the number of samples.
> > >
> > > We added the above discussion in Appendix A.1 of the revision paper.
> > >
> > > [1]. Shejwalkar, Virat, and Amir Houmansadr. "Manipulating the byzantine: Optimizing model poisoning attacks and defenses for federated learning." NDSS. 2021.
> > > [2]. Shejwalkar, Virat, et al. "Back to the drawing board: A critical evaluation of poisoning attacks on production federated learning." 2022 IEEE Symposium on Security and Privacy (SP). IEEE, 2022.
> > > [3]. Xie, Chulin, Minghao Chen, Pin-Yu Chen, and Bo Li. "Crfl: Certifiably robust federated learning against backdoor attacks." In International Conference on Machine Learning, pp. 11372-11382. PMLR, 2021.
> > > [4]. Wang, Xiaoyang, et al. "Invariant Aggregator for Defending Federated Backdoor Attacks." arXiv preprint arXiv:2210.01834 (2022).
> > > [5]. Nguyen, Thien Duc, et al. "{FLAME}: Taming Backdoors in Federated Learning." 31st USENIX Security Symposium (USENIX Security 22). 2022.
> > > [6]. Rieger, Phillip, et al. "Deepsight: Mitigating backdoor attacks in federated learning through deep model inspection." arXiv preprint arXiv:2201.00763 (2022).
> > > [7]. Tao, Guanhong, et al. "Model orthogonalization: Class distance hardening in neural networks for better security." 2022 IEEE Symposium on Security and Privacy (SP). IEEE. Vol. 3. 2022.
> > > [8]. Minka, Thomas. "Estimating a Dirichlet distribution." (2000): 4.
> > > [9]. Xie, Chulin, et al. "Dba: Distributed backdoor attacks against federated learning." International Conference on Learning Representations. 2019.
> > > [10]. Byzantine fault. https://en.wikipedia.org/wiki/Byzantine_fault.

---

> > > > ### Comment · Reviewer_zr2z · 2022-11-18
> > > > **Further comments about the rebuttal and the paper**
> > > >
> > > > Thanks for the rebuttal. I still have the following doubts:
> > > >
> > > >
> > > > {\bf Doubts about the threat model and algorithm design.}
> > > >
> > > > My understanding is that (based on Figure 1), before training, a defender already knows which are malicious clients, and which are benign clients. The last paragraph in A.1 also says that “”In each round we randomly select 10 clients, including 4 adversaries and 6 benign clients… Benign clients perform adversarial training continuously in both settings. “. How can it be possible that a defender know  benign and malicious in advance?   When checking Algorithm 1 GLOBAL MODEL INFERENCE, does that mean the server knows the trigger \delta already?
> > > > Can you elaborate more on how the proposed FLIP is trained and evaluated?
> > > >
> > > >
> > > > {\bf Doubts about the usefulness/practicability of the Theorems}
> > > >
> > > > In the proof of Theorem 1 (starting from Equation 9): Do all data samples and clients share the same recovered trigger ($\delta + \epsilon$)? Still, is it realistic?
> > > >
> > > > The proof is only applied to 2 clients in total and just shows the results for only 1 global round, instead of multiple rounds?
> > > >
> > > > In Theorem 2: $\alpha$:  is the numerator a scalar,  but the denominator a vector?
> > > >
> > > > I suggest the authors carefully proofread the proofs and make the assumptions and statements clear and accurate!
> > > >
> > > > What are the exact value of $\Delta min\_{loss}$  $\Delta max\_{loss}$ (between Theorem 2 and Corollary 1) in the considered 2 clients (1 benign + 1 malicious scenario) experiments? Should those values meaningful?
> > > >
> > > >
> > > > {\bf Doubts about the recovered trigger and defense.}
> > > >
> > > > The authors reply: “The triggers recovered by FLIP did not look exactly like the ground truth triggers but they shared common features”. If the recovered triggers are significantly different the true triggers, how about benign clients just randomly injecting noisy triggers locally (and flipping the labels) as a defense?

---

> > > > > ### Author Response · Authors · 2022-11-20
> > > > > **Official Follow-up Response to Reviewer zr2z (3/3)**
> > > > >
> > > > > > 11. {\bf Doubts about the recovered trigger and defense.}
> > > > >
> > > > > > 11.1 If the recovered triggers are significantly different the true triggers, how about benign clients just randomly injecting noisy triggers locally (and flipping the labels) as a defense?
> > > > >
> > > > > We speculate there might be a misunderstanding. In general, the inverted trigger will be slightly different from the ground truth trigger with respect to the shape and the location, but won’t be significantly different. They share a lot of common features. (See Appendix A.1 for example.)
> > > > >
> > > > > We conducted an experiment as the reviewer suggested in the first response, benign clients randomly injecting noisy triggers locally and setting the labels to ground truth labels as a defense. The results were in Q3 of the first response and also in Appendix A.15 Table 11. The method is not effective.
> > > > >
> > > > > As the reviewer suggested, we conduct another experiment, benign clients randomly injecting noisy triggers locally and flipping the labels to random labels as a defense. The method is still not effective. Basically, the benign clients are learning random noisy features with random labels, this will hurt accuracy, so this method won’t help in both reducing ASR and maintaining ACC. Results can be found in the below table.
> > > > >
> > > > > | **Single-shot Attack**                 | **CIFAR10** |       |
> > > > > |----------------------------------------|:-----------:|:-----:|
> > > > > |                                        |     ACC     |  ASR  |
> > > > > | **No Defense**                         | 77.52       | 80.46 |
> > > > > | **Random trigger + random flip label** | 10.59       | 83.02 |
> > > > >
> > > > >
> > > > >
> > > > >
> > > > >
> > > > > [1]. Bagdasaryan, E., et al. "How tobackdoor federated learning." arXiv preprint arXiv:1807.00459 (2018).
> > > > > [2]. Xie, Chulin, et al. "Dba: Distributed backdoor attacks against federated learning." International Conference on Learning Representations. 2019.
> > > > > [3]. Xie, Chulin, et al. "Crfl: Certifiably robust federated learning against backdoor attacks." International Conference on Machine Learning. PMLR, 2021.
> > > > > [4]. Wang, Xiaoyang, et al. "Invariant Aggregator for Defending Federated Backdoor Attacks." arXiv preprint arXiv:2210.01834 (2022).

---

> > > > > > ### Comment · Reviewer_zr2z · 2022-11-21
> > > > > > **Further Comments**
> > > > > >
> > > > > > Thanks for your clarifications on my algorithm and evaluation comments!
> > > > > >
> > > > > > The below are my further ones:
> > > > > >
> > > > > > “When benign local clients receive the global model, they apply trigger inversion techniques independently to recover the triggers based on their local data”.
> > > > > >
> > > > > > My doubt is: if the benign clients do not know the trigger information at all, how can he decide the location/size/value of the trigger?
> > > > > >
> > > > > >
> > > > > > “No, every client will use its own data to conduct trigger recovery independently.”
> > > > > >
> > > > > > This should be the way you conduct the experiments, but not the same in the proof.  See the appendix between Eq 8 and Eq 9, “In global server, define $\delta$ as the malicious clients generated trigger, $\delta + \epsilon$ as the benign clients generated trigger.“, where you fix $\epsilon$ for all benign clients.  This is contradictory with the threat model.
> > > > > >
> > > > > > "10.2 The proof is only applied to 2 clients in total and just shows the results for only 1 global round, instead of multiple rounds?
> > > > > > Yes, but our proof can be easily extended to multiple rounds and multiple clients. For example, to consider multiple rounds, we can use Appendix A.12 Equation 8 to extend.”
> > > > > >
> > > > > > I do not think this response is persuasive. Honestly, generalizing to multiple rounds or multiple clients is not easy, especially when different malicious or benign clients have different triggers $\delta$ or epsilon. In other words, what you prove is largely different with what you claim in Theorem 1. You should make this very accurate, as otherwise it is rather confusing!

---

> > > > > > > ### Author Response · Authors · 2022-11-21
> > > > > > > **Official Follow-up Response to Reviewer zr2z (2/2)**
> > > > > > >
> > > > > > > [1]. Wang, Bolun, et al. "Neural cleanse: Identifying and mitigating backdoor attacks in neural networks." 2019 IEEE Symposium on Security and Privacy (SP). IEEE, 2019.
> > > > > > > [2]. Liu, Yingqi, et al. "Abs: Scanning neural networks for back-doors by artificial brain stimulation." Proceedings of the 2019 ACM SIGSAC Conference on Computer and Communications Security. 2019.
> > > > > > > [3]. Guo, Wenbo, et al. "Tabor: A highly accurate approach to inspecting and restoring trojan backdoors in ai systems." arXiv preprint arXiv:1908.01763 (2019).
> > > > > > > [4]. Tao, Guanhong, et al. "Model orthogonalization: Class distance hardening in neural networks for better security." 2022 IEEE Symposium on Security and Privacy (SP). IEEE. Vol. 3. 2022.
> > > > > > > [5]. Wang, Ren, et al. "Practical detection of trojan neural networks: Data-limited and data-free cases." European Conference on Computer Vision. Springer, Cham, 2020.
> > > > > > > [6]. Shen, Guangyu, et al. "Backdoor scanning for deep neural networks through k-arm optimization." International Conference on Machine Learning. PMLR, 2021.

---

> > > > > > > > ### Comment · Reviewer_zr2z · 2022-11-22
> > > > > > > > **Reply to the Authors' Rebuttal**
> > > > > > > >
> > > > > > > > Thanks for the authors' rebuttal! This address my technical concerns. However, I am still thinking that the theoretical results are narrow. I will increase my score though.

---

> > > > > > > > > ### Author Response · Authors · 2022-11-22
> > > > > > > > > **Official Follow-up Response to Reviewer zr2z**
> > > > > > > > >
> > > > > > > > > We sincerely thank the reviewer for the discussion, which has improved our submission.

---

> > > > > > > ### Author Response · Authors · 2022-11-21
> > > > > > > **Official Follow-up Response to Reviewer zr2z (1/2)**
> > > > > > >
> > > > > > > We truly appreciate the reviewer's continuous help in improving our paper. We address the new questions and comments as follows. We also wonder if we had sufficiently addressed the reviewers’ prior questions/comments.
> > > > > > >
> > > > > > > > 12. My doubt is: if the benign clients do not know the trigger information at all, how can he decide the location/size/value of the trigger?
> > > > > > >
> > > > > > > We included detailed discussions on trigger inversion in Section 3 (Methodology). Trigger inversion [1, 2, 3, 4, 5, 6] does not require knowing the location/side/value of the trigger beforehand. They are derived by iterative gradient descent during inversion. In particular, for each client we leverage universal trigger inversion that aims to generate a trigger that can flip samples of all the classes (other than the target class) to the target class. The trigger generation starts from a random pattern and gradually forms a shape similar to the ground truth driven by the inversion loss function. Furthermore, there are many existing works [1, 2, 3, 4, 5, 6] about backdoor trigger inversion defenses that are assuming no knowledge of trigger information.
> > > > > > >
> > > > > > >
> > > > > > > > 13. This should be the way you conduct the experiments, but not the same in the proof. See the appendix between Eq 8 and Eq 9, “In global server, define $\delta$ as the malicious clients generated trigger,  $\delta + \epsilon$ as the benign clients generated trigger.“, where you fix $\epsilon$ for all benign clients. This is contradictory with the threat model.
> > > > > > >
> > > > > > > Our proof focuses on the two-client setting (one benign and the other malicious), and therefore the statement is correct in this setting.
> > > > > > > Our experiment setup is consistent with the threat model. In our experiment, during training, every client will use their own data to conduct trigger recovery independently. In Section 5 (Experiment), we empirically show that our defense gives outstanding performance against state-of-the-art backdoor attacks.
> > > > > > > Besides, as stated in Section 4 (Theoretical analysis), in order to simplify the analysis, our theoretical analysis assumes there is one global server and two clients: one benign and the other malicious. In each round, one benign local client has one $\delta + \epsilon$ in our theoretical analysis. We will add a remark in our proof that the threat model considered in our theoretical analysis is this two-client setup.
> > > > > > >
> > > > > > > > 14. I do not think this response is persuasive. Honestly, generalizing to multiple rounds or multiple clients is not easy, especially when different malicious or benign clients have different triggers $\delta$ or epsilon. In other words, what you prove is largely different with what you claim in Theorem 1. You should make this very accurate, as otherwise it is rather confusing!
> > > > > > >
> > > > > > > We are sorry that the reviewer finds our response confusing. We now better understand your concern, and will make it clearer that our current proof considers the two-client setting. In what follows, we would like to elaborate on how our theoretical framework can be extended to a more general setting. This discussion will also be included in the revised version (currently we are not allowed to update the manuscript).
> > > > > > >
> > > > > > > The extension can be considered in 3 folds.
> > > > > > >
> > > > > > > 1. two clients (including one benign and one malicious) & multiple rounds -> we can use Appendix A.12 Equation 8 to extend by changing the rounds $r$ and weights $W$.
> > > > > > > 2. multiple benign clients & (one malicious client) [or one backdoor trigger] -> we need to define different $\epsilon$ for each benign client and derive a new certificate following our proof in Appendix 11 and 12. Intuitively, the benign client having the worst trigger recovery performance will dominate the certificate.
> > > > > > > 3. multiple benign and malicious clients & (many malicious users with different triggers) ->  new analysis may be needed, and the definition of attack success needs to be reconsidered. As far as we know, no existing work (even in the non-FL backdoor defense literature) has provided theoretical analysis for such a complex setup. We leave developing this complex theoretical analysis for future work. One insight that could help the future work is that, say we have four malicious clients, though each client can inject different triggers, like “I”, “C” “L”, and “R”. However, in the end, the learned triggers will be one trigger, “ICLR”, in the model. As for the benign clients/defenders, they do not have to distinguish different sub-triggers, they can take what the global model learned as $\delta$. No matter how the malicious inject triggers, as long as the defender can invert the global model learned trigger, the defense will be considered successful.
> > > > > > >
> > > > > > > We will clarify these settings in the revised version to avoid confusion.

---

> > > > > ### Author Response · Authors · 2022-11-20
> > > > > **Official Follow-up Response to Reviewer zr2z (2/3)**
> > > > >
> > > > > > 10.3 In Theorem 2: $\alpha$: is the numerator a scalar, but the denominator a vector?
> > > > >
> > > > >
> > > > > Our derivation is correct. To make it clear, let us go through every dimensoin of the associated parameters.
> > > > >
> > > > > As mentioned in Section 4 (Theoretical Analysis), $ \mathbf{x} \in \mathbb{R}^{1 \times d_{x}}$, $\mathbf{z}$ has the same dimension as $ \mathbf{x}$, $\mathbf{z} \in \mathbb{R}^{1 \times d_{x}}$, $ \mathbf{q} \in \mathbb{R}^{1 \times I}$.
> > > > >
> > > > > In the numerator, $(\mathbf{q_{j}} - p(\mathbf{z_{j}}))$  is a vector with $1 \times I$ (dimensions),  $\mathbf{z_{j}}^{T}$ is a vector with $d \times 1$, then $[  \mathbf{z_{j}}^{T} (\mathbf{q_{j}} - p(\mathbf{z_{j}})) ]$ is a $d \times I$ matrix. $\mathbf{z_{s}}$ is a vector with $1 \times d$, then $ \mathbf{z_{s}} \sum_{j = 1}^{n_{1}}[  \mathbf{z_{j}}^{T} (\mathbf{q_{j}} - p(\mathbf{z_{j}})) ] $ is a $1 \times I$ vector, and we choose the $i$-th dimension and get a scalar. $q^{\*}{s, i}- q_{s, i}$ is a scalar (note that due to markdown compilation, the correct form is \q^{\*}\_{s, i}). So the numerator is a scalar.
> > > > >
> > > > > In the denominator, similar to part of numerator, $[  \mathbf{z_{j}}^{T} (\mathbf{q_{j}} - p(\mathbf{z_{j}})) ]$ is a $d \times I$ matrix, here we choose the $i$-th column, and get $d \times 1$ vector. $q^{\*}{s, i}- q_{s, i}$ is a scalar (note that due to markdown compilation, the correct form is \q^{\*}\_{s, i}).  $\mathbf{b}$ is a $1 \times d$ vector. Then in the denominator, we get $1 \times d$ multiply $d \times 1$ and get a scalar. So the denominator is also a scalar.
> > > > >
> > > > >
> > > > > > 10.4 What are the exact value of $\Delta \min\_loss$ $\Delta \max\_loss$  (between Theorem 2 and Corollary 1) in the considered 2 clients (1 benign + 1 malicious scenario) experiments? Should those values meaningful?
> > > > >
> > > > >
> > > > > As mentioned in our paper, Section 4 (Theoretical Analysis), $\Delta \min\_{loss}$ $\Delta \max\_{loss}$ are evaluated on the global server, it represents the global model **loss changes** on **backdoored** and **clean data** in the settings **with and without the defense**.
> > > > >
> > > > > We explained in our paper, “To facilitate the analysis of attack success rate (ASR) and clean accuracy (ACC) changes, intuitively, we aim to analyze how much the ASR at least will be reduced and how much the ACC will at most be maintained.”
> > > > >
> > > > > “Thus, we studied this lower bound ($\Delta \min\_{loss}$) on backdoor data, which indicates the minimal improvements on the backdoor defense that reduce the ASR.”
> > > > >
> > > > > “Similarly, we studied the upper bound ($\Delta \max\_{loss}$) for clean data, as they indicate the worst-case accuracy degradation.”
> > > > >
> > > > > These values are meaningful, by using the loss changes and the confidence threshold, we can exactly calculate both rejected backdoor samples with and without defense, and the rejected clean (benign) samples with and without defense, in order to have a holistic understanding of their effects. We conduct an experiment that follows the same setting as our assumptions in the theoretical analysis to validate its correctness. The experiment can be found in Section 5.3 (Evaluation on the Same Setting as Theoretical Analysis) and Table 3.
> > > > >
> > > > > As the reviewer suggested, we also plot a figure (https://imgur.com/a/nmuzI1K) to illustrate how $\Delta \min\_{loss}$ and $\Delta \max\_{loss}$ changes in each round. Note that the y-axis scale is different for clean data loss changes and poison data loss changes and we use a pretrained model with 30 rounds.
> > > > >
> > > > > As we explained in our paper, we consider $\Delta \max\_{loss}$ as the clean data loss changes, in order to maintain ACC, when applying defense $\Delta \max\_{loss}$ loss changes smaller will be better, we can see from the figure, the $\Delta \max\_{loss}$ changes at most within (0, 0.15).
> > > > >
> > > > > We consider $\Delta \min\_{loss}$ as the backdoor data loss changes, in order to reduce ASR, when applying defense $\Delta \min\_{loss}$ changes larger will be better, we can see from the figure, the $\Delta \min\_{loss}$ keep increasing when applying our defense.

---

> > > > > ### Author Response · Authors · 2022-11-20
> > > > > **Official Follow-up Response to Reviewer zr2z (1/3)**
> > > > >
> > > > > > 9. {\bf Doubts about the threat model and algorithm design.}
> > > > >
> > > > > >  9.1 My understanding is that (based on Figure 1), before training, a defender already knows which are malicious clients, and which are benign clients. How can it be possible that a defender know benign and malicious in advance?
> > > > >
> > > > > The defender does not have such knowledge.  As stated in Section 1 (Introduction: Threat Model), “On benign clients, we do not assume any knowledge about the ground truth trigger. Backdoor triggers are inverted on benign clients based on received model weights (from the global server) and their local data (non-i.i.d.).” We will add a note to the caption of Figure 1 to avoid confusion.
> > > > >
> > > > > First, during training, each benign client can be considered as a defender. As benign clients do not want the global model to be poisoned, they will choose to participate in the defense. Second, benign clients have no knowledge about the trigger and they have no knowledge about which are malicious clients either. Benign clients follow the FL protocol, and they do not communicate with each other. In each round, benign clients invert triggers based on their local data and the received global model. Third, as mentioned in our paper, the global server  has no knowledge about which client is malicious or benign.
> > > > >
> > > > > > 9.2 When checking Algorithm 1 GLOBAL MODEL INFERENCE, does that mean the server knows the trigger \delta already?
> > > > >
> > > > > No, the global server does not know the trigger, nor have the data. The Global Inference function was only for the evaluation purpose. In our algorithm, the global server’s input includes weights, confidence threshold and test samples (Algorithm 1 line 7). In order to avoid such a confusion, we have updated Algorithm 1 and removed the evaluation function.
> > > > >
> > > > >
> > > > > > 9.3 Can you elaborate more on how the proposed FLIP is trained and evaluated?
> > > > >
> > > > > As illustrated in Section 3 (Methodology), FLIP consists of three main steps, (1) Trigger inversion; (2) Model hardening; (3) Low-confidence sample rejection. During training, the global server distributes the global model to each client. When benign local clients receive the global model, they apply trigger inversion techniques independently to recover the triggers based on their local data. They do not know which are the malicious clients and do not have the trigger knowledge. Then benign local clients will combine the augmented data with the clean data to perform (local) model hardening and then submit the updated local model weights to the global server. The global server aggregates all the received weights, performs the inference and precludes samples with low prediction confidence.
> > > > >
> > > > > For FLIP evaluation details, as illustrated in Section 5 (Experiment), we evaluate FLIP under two existing attack settings[1, 2], and following their original settings. There are 100 clients in total by default. In each round, the global server randomly selects 10 clients, including 4 adversaries and 6 benign clients. We compare the performance of FLIP with 9 state-of-the-art defenses and we consider the attack success rate (ASR) and the main task accuracy (ACC) as the evaluation metrics to measure defense effectiveness. More details are in Appendix A.1 (Experiment Setup).
> > > > >
> > > > >
> > > > > > 10. {\bf Doubts about the usefulness/practicability of the Theorems}
> > > > >
> > > > > > 10.1 In the proof of Theorem 1 (starting from Equation 9): Do all data samples and clients share the same recovered trigger $(\delta+\epsilon)$? Still, is it realistic?
> > > > >
> > > > > No, every client will use its own data to conduct trigger recovery independently. This is a realistic setting, because in FL, clients can only access their own local data, and they will invert trigger based on the data they have.
> > > > >
> > > > > > 10.2 The proof is only applied to 2 clients in total and just shows the results for only 1 global round, instead of multiple rounds?
> > > > >
> > > > >
> > > > > Yes, but our proof can be easily extended to multiple rounds and multiple clients. For example, to consider multiple rounds, we can use Appendix A.12 Equation 8 to extend. Furthermore, existing FL backdoor works [3, 4] also performed theoretical analysis on one round.

---

> > > > > ### Comment · Reviewer_zr2z · 2022-11-23
> > > > > **Further Issues about the novelty / differences with the IEEE SP22**
> > > > >
> > > > > At first, I thought the backdoor inversion and model hardening techniques are proposed by the authors, which makes me think that the algorithm is novel. However, after I carefully read the Model Orthogonalization paper (Tao et al., IEEE SP22), I found these techniques are directly from that paper (for backdoor attacks to CV applications but can be easily generalized to other application domains such as NLP and learning paradigms such as FL) and simply applied to the FL setting (while the authors do not mention that paper details). Hence, I think the technical novelty is also limited (besides the limited theoretical results).  Due to this reason, I will still maintain the "reject" decision.  Please point out and discuss it if what I said is inaccurate!

---

> > > > > > ### Author Response · Authors · 2022-11-24
> > > > > > **Official Follow-up Response to Reviewer zr2z (2/2)**
> > > > > >
> > > > > > [1]. Kurakin, Alexey, Ian Goodfellow, and Samy Bengio. "Adversarial machine learning at scale." arXiv preprint arXiv:1611.01236 (2016).
> > > > > > [2]. Madry, Aleksander, et al. "Towards deep learning models resistant to adversarial attacks." arXiv preprint arXiv:1706.06083 (2017).
> > > > > > [3]. Tramèr, Florian, et al. "Ensemble adversarial training: Attacks and defenses." arXiv preprint arXiv:1705.07204 (2017).
> > > > > > [4]. Moosavi-Dezfooli, Seyed-Mohsen, et al. "Universal adversarial perturbations." Proceedings of the IEEE conference on computer vision and pattern recognition. 2017.
> > > > > > [5]. Shafahi, Ali, et al. "Adversarial training for free!." Advances in Neural Information Processing Systems 32 (2019).
> > > > > > [6]. Wong, Eric, Leslie Rice, and J. Zico Kolter. "Fast is better than free: Revisiting adversarial training." arXiv preprint arXiv:2001.03994 (2020).
> > > > > > [7]. Tao, Guanhong, et al. "Model orthogonalization: Class distance hardening in neural networks for better security." 2022 IEEE Symposium on Security and Privacy (SP). IEEE. Vol. 3. 2022.
> > > > > > [8]. Shejwalkar, Virat, and Amir Houmansadr. "Manipulating the byzantine: Optimizing model poisoning attacks and defenses for federated learning." NDSS. 2021.
> > > > > > [9]. Liu, Yingqi, et al. "Abs: Scanning neural networks for back-doors by artificial brain stimulation." Proceedings of the 2019 ACM SIGSAC Conference on Computer and Communications Security. 2019.
> > > > > > [10]. Zhang, Hongyang, et al. "Theoretically principled trade-off between robustness and accuracy." International conference on machine learning. PMLR, 2019.
> > > > > > [11]. Lin, Tao, et al. "Ensemble distillation for robust model fusion in federated learning." Advances in Neural Information Processing Systems 33 (2020): 2351-2363.
> > > > > > [12]. Aramoon, Omid, et al. "Meta Federated Learning." arXiv preprint arXiv:2102.05561 (2021).
> > > > > > [13]. Li, Suyi, et al. "Learning to detect malicious clients for robust federated learning." arXiv preprint arXiv:2002.00211 (2020).
> > > > > > [14]. Carlini, Nicholas, and David A. Wagner. "Towards evaluating the robustness of neural networks. corr abs/1608.04644 (2016)." arXiv preprint arXiv:1608.04644 (2016).
> > > > > > [15]. Goodfellow, Ian J., Jonathon Shlens, and Christian Szegedy. "Explaining and harnessing adversarial examples." arXiv preprint arXiv:1412.6572 (2014).

---

> > > > > > ### Author Response · Authors · 2022-11-24
> > > > > > **Official Follow-up Response to Reviewer zr2z (1/2)**
> > > > > >
> > > > > > > 15. At first, I thought the backdoor inversion and model hardening techniques are proposed by the authors, which makes me think that the algorithm is novel. However, after I carefully read the Model Orthogonalization paper (Tao et al., IEEE SP22), I found these techniques are directly from that paper (for backdoor attacks to CV applications but can be easily generalized to other application domains such as NLP and learning paradigms such as FL) and simply applied to the FL setting (while the authors do not mention that paper details). Hence, I think the technical novelty is also limited (besides the limited theoretical results). Due to this reason, I will still maintain the "reject" decision. Please point out and discuss it if what I said is inaccurate!
> > > > > >
> > > > > > We appreciate the reviewer’s continuous help.
> > > > > > However, we respectfully disagree with the statement that our paper is a direct application of SP22 [7] to the FL setting. In fact, we never hid that our work leveraged part of SP22. In each place that we utilized any algorithm in SP22, we clearly indicated so (e.g., in Section 3). More importantly, FLIP goes well beyond SP22 by addressing a number of critical challenges in the FL setting. Our description of the paper’s contributions in Section 1 (Introduction) precisely reflected this.
> > > > > >
> > > > > > We elaborate the differences between FLIP and SP22 in detail as follows.
> > > > > >
> > > > > > 1. As Reviewer zr2z mentioned in the first phase rebuttal Q5, “Non-IID is an important factor that affects the defense performance.” Non-IID is a critical problem in FL. We propose a novel cached warm-up for each benign local client, which is specially designed for federated learning (not in SP22). When a client is selected again, there is no need to conduct the warm-up, which can significantly improve the efficiency of trigger inversion on local clients. Such a design is newly proposed in our paper and was not in SP22 paper.
> > > > > > 2. Due to the Non-IID nature in federated learning, the symmetric hardening method proposed in SP22 (one of its major contributions) is no longer feasible under the FL setting. We propose a novel method using both asymmetric and symmetric hardening on benign local clients, depending on their local data availability. A direct application of SP22 cannot handle Non-IID settings. In addition, as mentioned by Reviewer zr2z, SOTA baseline DnC [8] can only work under IID and is ineffective under Non-IID. FLIP surpasses all existing SOTA baselines.
> > > > > > 3. Besides the above two differences from SP22, we also introduce a new low-confidence threshold rejection procedure, which is only feasible in the FL setting. Specifically, we apply an additional sample filtering step, in which we use a threshold to preclude samples with low prediction confidence.
> > > > > > 4. Furthermore, we have empirically demonstrated that FLIP is still effective under much stronger adaptive attacks, whereas SP22 does not consider such strong attacks (details in Appendix A.3). We also mentioned this in our contributions in Section 1 (Introduction: Contributions).
> > > > > > 5. SP22 did not have any theoretical analysis. Our theoretical analysis is new and specialized in the FL setting.
> > > > > >
> > > > > > We further argue that in the context of FL backdoor defense, existing methods mainly focus on inspecting model weights and often require extra data [11, 12, 13]. In contrast, FLIP provides a completely new perspective based on model hardening.
> > > > > >
> > > > > > Note that in the classic adversarial training area, a large corpus of techniques e.g., [3,4,5,6,10], were developed (as extensions to various scenarios) after the first few seminal works were proposed such as C&W [14], PGD [2], and FGSM [15]. These extensions made important contributions in various application areas. Similarly, the existence of SP22 should not stop further development of model hardening techniques.

---

> > > > ### Author Response · Authors · 2022-11-20
> > > > **Official Follow-up Response to Reviewer zr2z (3/3)**
> > > >
> > > > > 11. {\bf Doubts about the recovered trigger and defense.}
> > > >
> > > > > 11.1 If the recovered triggers are significantly different the true triggers, how about benign clients just randomly injecting noisy triggers locally (and flipping the labels) as a defense?
> > > >
> > > > We speculate there might be a misunderstanding. In general, the inverted trigger will be slightly different from the ground truth trigger with respect to the shape and the location, but won’t be significantly different. They share a lot of common features. (See Appendix A.1 for example.)
> > > >
> > > > We conducted an experiment as the reviewer suggested in the first response, benign clients randomly injecting noisy triggers locally and setting the labels to ground truth labels as a defense. The results were in Q3 of the first response and also in Appendix A.15 Table 11. The method is not effective.
> > > >
> > > > As the reviewer suggested, we conduct another experiment, benign clients randomly injecting noisy triggers locally and flipping the labels to random labels as a defense. The method is still not effective. Basically, the benign clients are learning random noisy features with random labels, this will hurt accuracy, so this method won’t help in both reducing ASR and maintaining ACC. Results can be found in the below table.
> > > >
> > > > | **Single-shot Attack**                 | **CIFAR10** |       |
> > > > |----------------------------------------|:-----------:|:-----:|
> > > > |                                        |     ACC     |  ASR  |
> > > > | **No Defense**                         | 77.52       | 80.46 |
> > > > | **Random trigger + random flip label** | 10.59       | 83.02 |
> > > >
> > > >
> > > >
> > > >
> > > >
> > > > [1]. Bagdasaryan, E., et al. "How tobackdoor federated learning." arXiv preprint arXiv:1807.00459 (2018).
> > > > [2]. Xie, Chulin, et al. "Dba: Distributed backdoor attacks against federated learning." International Conference on Learning Representations. 2019.
> > > > [3]. Xie, Chulin, et al. "Crfl: Certifiably robust federated learning against backdoor attacks." International Conference on Machine Learning. PMLR, 2021.
> > > > [4]. Wang, Xiaoyang, et al. "Invariant Aggregator for Defending Federated Backdoor Attacks." arXiv preprint arXiv:2210.01834 (2022).

---

> > > > ### Author Response · Authors · 2022-11-20
> > > > **Official Follow-up Response to Reviewer zr2z (2/3)**
> > > >
> > > > > 10.3 In Theorem 2: $\alpha$: is the numerator a scalar, but the denominator a vector?
> > > >
> > > >
> > > > Our derivation is correct. To make it clear, let us go through every dimensoin of the associated parameters.
> > > >
> > > > As mentioned in Section 4 (Theoretical Analysis), $ \mathbf{x} \in \mathbb{R}^{1 \times d_{x}}$, $\mathbf{z}$ has the same dimension as $ \mathbf{x}$, $\mathbf{z} \in \mathbb{R}^{1 \times d_{x}}$, $ \mathbf{q} \in \mathbb{R}^{1 \times I}$.
> > > >
> > > > In the numerator, $(\mathbf{q_{j}} - p(\mathbf{z_{j}}))$  is a vector with $1 \times I$ (dimensions),  $\mathbf{z_{j}}^{T}$ is a vector with $d \times 1$, then $[  \mathbf{z_{j}}^{T} (\mathbf{q_{j}} - p(\mathbf{z_{j}})) ]$ is a $d \times I$ matrix. $\mathbf{z_{s}}$ is a vector with $1 \times d$, then $ \mathbf{z_{s}} \sum_{j = 1}^{n_{1}}[  \mathbf{z_{j}}^{T} (\mathbf{q_{j}} - p(\mathbf{z_{j}})) ] $ is a $1 \times I$ vector, and we choose the $i$-th dimension and get a scalar. $q^{\*}{s, i}- q_{s, i}$ is a scalar (note that due to markdown compilation, the correct form is \q^{\*}\_{s, i}). So the numerator is a scalar.
> > > >
> > > > In the denominator, similar to part of numerator, $[  \mathbf{z_{j}}^{T} (\mathbf{q_{j}} - p(\mathbf{z_{j}})) ]$ is a $d \times I$ matrix, here we choose the $i$-th column, and get $d \times 1$ vector. $q^{\*}{s, i}- q_{s, i}$ is a scalar (note that due to markdown compilation, the correct form is \q^{\*}\_{s, i}).  $\mathbf{b}$ is a $1 \times d$ vector. Then in the denominator, we get $1 \times d$ multiply $d \times 1$ and get a scalar. So the denominator is also a scalar.
> > > >
> > > >
> > > > > 10.4 What are the exact value of $\Delta \min\_loss$ $\Delta \max\_loss$  (between Theorem 2 and Corollary 1) in the considered 2 clients (1 benign + 1 malicious scenario) experiments? Should those values meaningful?
> > > >
> > > >
> > > > As mentioned in our paper, Section 4 (Theoretical Analysis), $\Delta \min\_{loss}$ $\Delta \max\_{loss}$ are evaluated on the global server, it represents the global model **loss changes** on **backdoored** and **clean data** in the settings **with and without the defense**.
> > > >
> > > > We explained in our paper, “To facilitate the analysis of attack success rate (ASR) and clean accuracy (ACC) changes, intuitively, we aim to analyze how much the ASR at least will be reduced and how much the ACC will at most be maintained.”
> > > >
> > > > “Thus, we studied this lower bound ($\Delta \min\_{loss}$) on backdoor data, which indicates the minimal improvements on the backdoor defense that reduce the ASR.”
> > > >
> > > > “Similarly, we studied the upper bound ($\Delta \max\_{loss}$) for clean data, as they indicate the worst-case accuracy degradation.”
> > > >
> > > > These values are meaningful, by using the loss changes and the confidence threshold, we can exactly calculate both rejected backdoor samples with and without defense, and the rejected clean (benign) samples with and without defense, in order to have a holistic understanding of their effects. We conduct an experiment that follows the same setting as our assumptions in the theoretical analysis to validate its correctness. The experiment can be found in Section 5.3 (Evaluation on the Same Setting as Theoretical Analysis) and Table 3.
> > > >
> > > > As the reviewer suggested, we also plot a figure (https://imgur.com/a/nmuzI1K) to illustrate how $\Delta \min\_{loss}$ and $\Delta \max\_{loss}$ changes in each round. Note that the y-axis scale is different for clean data loss changes and poison data loss changes and we use a pretrained model with 30 rounds.
> > > >
> > > > As we explained in our paper, we consider $\Delta \max\_{loss}$ as the clean data loss changes, in order to maintain ACC, when applying defense $\Delta \max\_{loss}$ loss changes smaller will be better, we can see from the figure, the $\Delta \max\_{loss}$ changes at most within (0, 0.15).
> > > >
> > > > We consider $\Delta \min\_{loss}$ as the backdoor data loss changes, in order to reduce ASR, when applying defense $\Delta \min\_{loss}$ changes larger will be better, we can see from the figure, the $\Delta \min\_{loss}$ keep increasing when applying our defense.

---

> > > > ### Author Response · Authors · 2022-11-20
> > > > **Official Follow-up Response to Reviewer zr2z (1/3)**
> > > >
> > > > > 9. {\bf Doubts about the threat model and algorithm design.}
> > > >
> > > > >  9.1 My understanding is that (based on Figure 1), before training, a defender already knows which are malicious clients, and which are benign clients. How can it be possible that a defender know benign and malicious in advance?
> > > >
> > > > The defender does not have such knowledge.  As stated in Section 1 (Introduction: Threat Model), “On benign clients, we do not assume any knowledge about the ground truth trigger. Backdoor triggers are inverted on benign clients based on received model weights (from the global server) and their local data (non-i.i.d.).” We will add a note to the caption of Figure 1 to avoid confusion.
> > > >
> > > > First, during training, each benign client can be considered as a defender. As benign clients do not want the global model to be poisoned, they will choose to participate in the defense. Second, benign clients have no knowledge about the trigger and they have no knowledge about which are malicious clients either. Benign clients follow the FL protocol, and they do not communicate with each other. In each round, benign clients invert triggers based on their local data and the received global model. Third, as mentioned in our paper, the global server  has no knowledge about which client is malicious or benign.
> > > >
> > > > > 9.2 When checking Algorithm 1 GLOBAL MODEL INFERENCE, does that mean the server knows the trigger \delta already?
> > > >
> > > > No, the global server does not know the trigger, nor have the data. The Global Inference function was only for the evaluation purpose. In our algorithm, the global server’s input includes weights, confidence threshold and test samples (Algorithm 1 line 7). In order to avoid such a confusion, we have updated Algorithm 1 and removed the evaluation function.
> > > >
> > > >
> > > > > 9.3 Can you elaborate more on how the proposed FLIP is trained and evaluated?
> > > >
> > > > As illustrated in Section 3 (Methodology), FLIP consists of three main steps, (1) Trigger inversion; (2) Model hardening; (3) Low-confidence sample rejection. During training, the global server distributes the global model to each client. When benign local clients receive the global model, they apply trigger inversion techniques independently to recover the triggers based on their local data. They do not know which are the malicious clients and do not have the trigger knowledge. Then benign local clients will combine the augmented data with the clean data to perform (local) model hardening and then submit the updated local model weights to the global server. The global server aggregates all the received weights, performs the inference and precludes samples with low prediction confidence.
> > > >
> > > > For FLIP evaluation details, as illustrated in Section 5 (Experiment), we evaluate FLIP under two existing attack settings[1, 2], and following their original settings. There are 100 clients in total by default. In each round, the global server randomly selects 10 clients, including 4 adversaries and 6 benign clients. We compare the performance of FLIP with 9 state-of-the-art defenses and we consider the attack success rate (ASR) and the main task accuracy (ACC) as the evaluation metrics to measure defense effectiveness. More details are in Appendix A.1 (Experiment Setup).
> > > >
> > > >
> > > > > 10. {\bf Doubts about the usefulness/practicability of the Theorems}
> > > >
> > > > > 10.1 In the proof of Theorem 1 (starting from Equation 9): Do all data samples and clients share the same recovered trigger $(\delta+\epsilon)$? Still, is it realistic?
> > > >
> > > > No, every client will use its own data to conduct trigger recovery independently. This is a realistic setting, because in FL, clients can only access their own local data, and they will invert trigger based on the data they have.
> > > >
> > > > > 10.2 The proof is only applied to 2 clients in total and just shows the results for only 1 global round, instead of multiple rounds?
> > > >
> > > >
> > > > Yes, but our proof can be easily extended to multiple rounds and multiple clients. For example, to consider multiple rounds, we can use Appendix A.12 Equation 8 to extend. Furthermore, existing FL backdoor works [3, 4] also performed theoretical analysis on one round.

---

### Official Review · Reviewer_cMnB · 2022-10-23

**Confidence:** 5
**Correctness:** 4
**Technical Novelty And Significance:** 4
**Empirical Novelty And Significance:** 4
**Recommendation:** 8

**Clarity, Quality, Novelty And Reproducibility:**

The paper is well written and easy to understand.
They claim to open source the code.

**Strength And Weaknesses:**

Strengths:
The underlying FLIP algorithm is rather novel. It combines existing concepts but in an interesting way.
The theoretical analysis is also interesting. The statements state results that are definitely relevant and intriguing to show.

Weaknesses:
Perhaps FLIP can be enhanced to improve its performance. It would be great if it beats other benchmarks in more cases.


**Summary Of The Paper:**

In FL many clients participate. Malign/adversary clients can potentially join and impact training by submitting malicious weights. To cope with such scenarios benign clients can employ algorithms that are more robust for such injections. The authors propose the use of data augmentation through adversarial training that benign clients should use.
The also provide a theoretical analysis of the difference in the loss function if benign clients employ such as strategy vs not.
On standard datasets they find that their algorithm usually outperforms benchmark algorithms.

**Summary Of The Review:**

The contributions are significant. The paper is a nice mixture of algorithmic designs and theoretical analyses.

The literature review section can be improved by more explicitly stating how does prior work differ from the current work.

---

> ### Author Response · Authors · 2022-11-17
> **Official Response to Reviewer cMnB**
>
> > 1. Perhaps FLIP can be enhanced to improve its performance. It would be great if it beats other benchmarks in more cases.
>
> We thank the reviewer for the valuable suggestion. We compared with another SOTA defense baseline DnC [1] and now we have compared with nine SOTA defense techniques in Section 5 (Experiment). Compared with DnC, FLIP performs better in single-shot attack settings and significantly outperforms in continuous attack settings. Under continuous attack settings, in MNSIT, FLIP reduces the ASR to 1.93% and keeps the ACC at 96.62%, while DnC’s ASR is 99.78% and its ACC drops to 89.81%; in Fashion MNIST, FLIP reduces the ASR to 17.65% and keeps the ACC at 72.99%, while DnC’s ASR is 97.73%, and its ACC drops to 65.90%; in CIFAR-10, FLIP reduces the ASR to 22.90% and keeps the ACC at 71.28%, while DnC’s ASR is 82.37% and its ACC drops to 53.16%. More in Table 1 and Table 2.
>
> We have discussed the trigger performance under injecting random triggers and ground truth triggers in Appendix A.15 Table 11, and also discussed the Non-IID factor effects in Appendix A.16 Table 12. Observed from Appendix A.15, comparing FLIP with ground truth trigger performance, there still have improvement space in the trigger inversion part in FLIP. We leave further improving FLIP performance for future work.
>
> > 2. The literature review section can be improved by more explicitly stating how does prior work differ from the current work.
>
> We thank the reviewer for the constructive suggestions. We discussed more how the existing works differ from ours in Section 2 (Related Work). We also discussed the deficiencies of existing defenses in Section 1 (Introduction) and we have a section in Appendix A.11 that discusses several possible reasons why existing defenses do not work well.
>
> [1]. Shejwalkar, Virat, and Amir Houmansadr. "Manipulating the byzantine: Optimizing model poisoning attacks and defenses for federated learning." NDSS. 2021.

---

### Official Review · Reviewer_cwn7 · 2022-10-25

**Confidence:** 3
**Clarity, Quality, Novelty And Reproducibility:** The authors will publish codes after …
**Correctness:** 3
**Technical Novelty And Significance:** 3
**Empirical Novelty And Significance:** 3
**Recommendation:** 6

**Strength And Weaknesses:**

1. The paper presents a new provable defense framework that can provide a sufficient condition on the quality of trigger recovery such that the proposed defense is provably effective in mitigating backdoor attacks.
2. The paper empirically evaluate the effectiveness of FLIP at scale across MNIST, Fashion-MNIST and CIFAR-10, using non-linear neural networks. The results show that FLIP significantly outperforms SOTAs on the continuous FL backdoor attack setting.




**Summary Of The Paper:**

The paper propose a trigger reverse engineering based defense and show that the method can achieve robustness improvement with guarantee (i.e., reducing the attack success rate) without affecting benign accuracy. The paper conducts comprehensive experiments across different datasets and attack settings. The results on eight competing SOTA defense methods show the empirical superiority of the  method on both single-shot and continuous FL backdoor attacks.

**Summary Of The Review:**

See the above.

---

> ### Author Response · Authors · 2022-11-17
> **Official Response to Reviewer cwn7**
>
> > The paper presents a new provable defense framework that can provide a sufficient condition on the quality of trigger recovery and empirically evaluate the effectiveness of FLIP at scale across MNIST, Fashion-MNIST, and CIFAR-10, using non-linear neural networks.
>
> We thank the reviewer for the insightful comment and we added evaluations on the different factors affecting trigger quality, including different trigger shapes,  colors, and ground-truth trigger effects. More in Table 11.
>
> We added evaluations on different non-i.i.d. distributions to evaluate FLIP’s performance. More in Table 12.
>
> We also added another defense baseline (Dnc [1]) on both single-shot and continuous attack evaluations. Compared with DnC, FLIP performs better in single-shot attack settings and significantly outperforms in continuous attack settings. More in Table 1 and Table 2.
>
> [1]. Shejwalkar, Virat, and Amir Houmansadr. "Manipulating the byzantine: Optimizing model poisoning attacks and defenses for federated learning." NDSS. 2021.

---

### Official Review · Reviewer_i7z3 · 2022-10-27

**Confidence:** 4
**Clarity, Quality, Novelty And Reproducibility:** Please see the above comments.
**Correctness:** 4
**Technical Novelty And Significance:** 3
**Empirical Novelty And Significance:** 3
**Recommendation:** 8

**Strength And Weaknesses:**

Strength

+ This paper proposes a new federated learning method to defend against backdoor attacks. Instead of designing new aggregation rules or changing how the server aggregates local model updates, the method modifies how benign clients train their local models.

+ The paper extends centralized learning backdoor defense methods to federated learning. This is an interesting direction to explore.

+ Some theoretical analysis is performed.

+ Evaluation is extensive, covering multiple datasets, multiple baselines, and adaptive attacks.

Weakness

- One potential weakness is that the meaning of provable is slightly different from what provable defense usually means. My first impression about provable defense is that it provides certified accuracy, like what ensemble federated learning provides. But this is a minor clarification issue.

Other minor comment

Can you describe the setting for FLTrust? In particular, what root dataset is used by the server in FLTrust?



**Summary Of The Paper:**

Federated learning is vulnerable to backdoor attacks. The authors propose a new defense against backdoor attacks to federated learning. The key idea is to combine some techniques developed to defend against backdoor attacks to centralized learning. Some theoretical analysis is performed to analyze the loss and learnt parameters. Evaluation is extensive, covering both existing and adaptive attacks.

**Summary Of The Review:**

This paper extends centralized learning backdoor defense to federated learning, which is an interesting direction. Extensive evaluation is performed to show the effectiveness of the method.

---

> ### Author Response · Authors · 2022-11-17
> **Official Response to Reviewer i7z3**
>
> > 1. One potential weakness is that the meaning of provable is slightly different from what provable defense usually means. But this is a minor clarification issue.
>
> We thank the reviewer for bringing up the interesting point. We discussed this in the introduction and experiment section. Certified accuracy is commonly used in evasion attacks that do not involve training. As data poisoning happens during training, it is more reasonable to certify the behavior of models during training rather than inference. We will make this clear in the updated version.
>
>
> > 2. Can you describe the setting for FLTrust? In particular, what root dataset is used by the server in FLTrust?
>
>
> In FLTrust, we follow the original settings in [1] and collect the root dataset for the learning task with 100 training examples, the root dataset has the same distribution as the overall training data distribution of the learning task. We exclude the sampled root dataset from the clients’ local training data, indicating that the root dataset is collected independently by the global server.
>
> We have updated Appendix A.1 (Experiment Setup) of the revision to include these references and discussions of FLTrust settings.
>
> [1]. Cao, Xiaoyu, Minghong Fang, Jia Liu, and Neil Zhenqiang Gong. "Fltrust: Byzantine-robust federated learning via trust bootstrapping.” NDSS 2021.

---

### Author Response · Authors · 2022-11-17
**Paper Revision Summary**

We thank all the reviewers for their insightful questions and suggestions! We are glad that the reviewers found our paper studies an “important problem”, “well written and easy to understand”, “contributions are significant”, “ theoretical analysis is also interesting”, “evaluation is extensive”, and “interesting direction”.
Below is a summary of major paper updates:
1. [Abstract] Include another baseline, change from “eight” to “nine” baselines.
2. [Section 2] Include more related work and discussion on differences, following Reviewer cMnB’s, and Reviewer zr2z’s suggestions.
3. [Section 4] Correct symbols and add more illustrations on notations.
4. [Section 5] Add another defense baseline (Dnc [1]) on both single-shot and continuous attack evaluations.
5. [Appendix A.1] Provide discussions on one of the defense baseline FLTrust settings and provide details on the augmented dataset and class distance.
6. [Appendix A.12] Discuss more on linear model theoretical generalization and correct the case in generalization form.
7. [Appendix A.13] Correct symbols and provide more illustrations on notations.
8. [Appendix A.15] Add evaluations on the impact on trigger quality, study different shape and color triggers, and ground-truth trigger effects.
9. [Appendix A.16] Add evaluations on different non-i.i.d. distributions to evaluate FLIP performance.

Please also let us know if there are other questions, and we really look forward to the discussion with the reviewers to further improve our paper. Thank you!!

[1]. Shejwalkar, Virat, and Amir Houmansadr. "Manipulating the byzantine: Optimizing model poisoning attacks and defenses for federated learning." NDSS. 2021.

---

### Decision · Program_Chairs · 2023-01-20

**Decision:**

Accept: poster

**Justification For Why Not Higher Score:**

The scope of the paper is perhaps limited, both in terms of audience, but also in terms of applicability of results within that community of ICLR.

**Justification For Why Not Lower Score:**

The issue of defenses against backdoor attacks is quite important, and few provable defenses exist. This work contributes towards that direction in a non-trivial way.

**Metareview: Summary, Strengths And Weaknesses:**

The authors of this paper analyze the connection between cross-entropy loss, attack success rate, and clean accuracy in a Federated Learning (FL) setting. They also propose a trigger reverse engineering based defense to improve robustness without affecting benign accuracy. Finally, they conduct experiments across different datasets and attack settings to show empirically that their method is superior to eight existing methods for single-shot and continuous FL backdoor attacks.

The Reviewers identified the following list of strengths and weaknesses.

Strengths:
- Proposes a new federated learning method to defend against backdoor attacks
- Extends centralized learning backdoor defense methods to federated learning
- Performs some theoretical analysis
- Extensive evaluation covering multiple datasets, multiple baselines, and adaptive attacks
- Empirically evaluated the effectiveness of FLIP at scale across MNIST, Fashion-MNIST and CIFAR-10
- Significantly outperformed SOTAs on continuous FL backdoor attack setting

Weaknesses:
- Meaning of "provable" is slightly different from what is usually meant by "provable defense"
- Minor issues in the paper’s claims
- Statements not well-supported
- Requires small changes to be made correct
- Theoretical results contained some inaccuracies, and may not provide an accurate set of guidelines as to when this method works.

The most critical of reviewers engaged significantly with the authors during rebuttal, during which most technical concerns were lifted. Although the reviewer raised the score, that remained relatively low. However the justification was mostly on the basis of how big the scope of the defense is.

Overall this is an interesting paper, that combines a few novel ideas, and provides some non-trivial theory for a problem of interest in the FL community. Even if the defenses presented are not general enough, they may spark some discussion, and further idea generation on this important topic of FL robustness against backdoor attacks.


**Note From Pc:**

if the above contains the word "oral" or "spotlight" please see: "oral" presentation means -> notable-top-5% and "spotlight" means -> notable-top-25%. As stated in our emails, we are disassociating presentation type from AC recommendations

**Summary Of Ac-Reviewer Meeting:**

We were not able to sync schedules for a timely meeting.